# *N*-Acryloylindole-alkyne (NAIA) enables imaging and profiling new ligandable cysteines and oxidized thiols by chemoproteomics

Tin-Yan Koo[1,6], Hinyuk Lai[1,6], Daniel K. Nomura [2,3] & Clive Yik-Sham Chung [1,4,5] ✉

Cysteine has been exploited as the binding site of covalent drugs. Its high sensitivity to oxidation is also important for regulating cellular processes. To identify new ligandable cysteines which can be hotspots for therapy and to better study cysteine oxidations, we develop cysteine-reactive probes, *N*-acryloylindole-alkynes (NAIAs), which have superior cysteine reactivity owing to delocalization of π electrons of the acrylamide warhead over the whole indole scaffold. This allows NAIAs to probe functional cysteines more effectively than conventional iodoacetamide-alkyne, and to image oxidized thiols by confocal fluorescence microscopy. In mass spectrometry experiments, NAIAs successfully capture new oxidized cysteines, as well as a new pool of ligandable cysteines and proteins. Competitive activity-based protein profiling experiments further demonstrate the ability of NAIA to discover lead compounds targeting these cysteines and proteins. We show the development of NAIAs with activated acrylamide for advancing proteome-wide profiling and imaging ligandable cysteines and oxidized thiols.

Cysteine is one of the most intriguing amino acids due to its intrinsically high nucleophilicity and sensitivity to oxidation[1–8]. It can govern and regulate a variety of important biological processes[1]. Activity-based protein profiling (ABPP) has been demonstrated as a powerful platform for proteome-wide profiling of functional cysteines[2,3,6,7,9–16]. Interestingly, many of these cysteines liganded by activity-based probes in ABPP experiments are found to associate with disease development and propagation. This facilitates further study on using electrophiles, with acrylamide[6,12,17–22] and chloroacetamide[2,6,12,19,21,23] as notable examples, to target new ligandable hotspots for therapy. Therefore, the activity-based probe is important and can aid in the identification of new druggable hotspots and new therapeutic covalent ligands.

Examples of activity-based probes for cysteine include maleimides[2,24–26], α,β-unsaturated ketones[2,6,13,18,19,22,27] and haloacetamides[2–4,6,7,12,13,15,17,19,20,23,24,28], with iodoacetamide alkyne (IAA)[2–4,6,7,12–15,17,20,23,24,28–30] as the most widely used probe for proteome-wide cysteine profiling. Nonetheless, challenges in using IAA for cysteine profiling have been encountered, including side reactions with other nucleophilic amino acids such as lysine and serine, as well as oxidized thiols such as sulfenic acid[31,32]. Also, a large excess of IAA is required for good labeling of functional cysteines due to its relatively low reactivity. This limits its application for cell lysates labeling only but not live cells due to high toxicity from the high working concentration of IAA, in addition to its low biostability (Fig. 1)[33]. As a result,

[1]School of Biomedical Sciences, Li Ka Shing Faculty of Medicine, The University of Hong Kong, Hong Kong, P. R. China. [2]Department of Chemistry, University of California, Berkeley, Berkeley, CA, USA. [3]Department of Molecular and Cell Biology, University of California, Berkeley, Berkeley, CA, USA. [4]Department of Pathology, School of Clinical Medicine, Li Ka Shing Faculty of Medicine, The University of Hong Kong, Hong Kong, P. R. China. [5]Centre for Oncology and Immunology, Hong Kong Science Park, Hong Kong, P. R. China. [6]These authors contributed equally: Tin-Yan Koo, Hinyuk Lai ✉e-mail: cyschung@hku.hk

the development of new cysteine-reactive compounds as better cysteine probes[2,3,32,34–42] is one of the active research areas to advance ABPP and other experiments for studying cysteine biology.

We are interested in using acrylamide compounds as cysteine-reactive probes because of their known higher biostability than iodoacetamides[43], as well as excellent selectivity for the thiol group on cysteine through thiol-Michael addition reaction[17,18,22]. Yet, low cysteine reactivity has been found in aqueous buffer solutions for acrylamide compounds, in comparison to iodoacetamides[19]. To develop acrylamide compounds into potential cysteine-reactive

probes, we sought to activate the acrylamides with higher electrophilicity, so that they should react more readily with nucleophilic cysteines. This should be feasible by incorporating the acrylamide group with less electron-rich nitrogen. Indole nitrogen, with its lone pair π electrons delocalized over the whole indole molecule due to aromaticity, is a suitable scaffold for acrylamide incorporation and activation.

In this work, we report a class of cysteine-reactive compounds, *N*-acryloylindoles (NAIs) and *N*-acryloylindole-alkynes (NAIAs), which contain acrylamide warhead on the indole nitrogen (Fig. 1). NAIA is

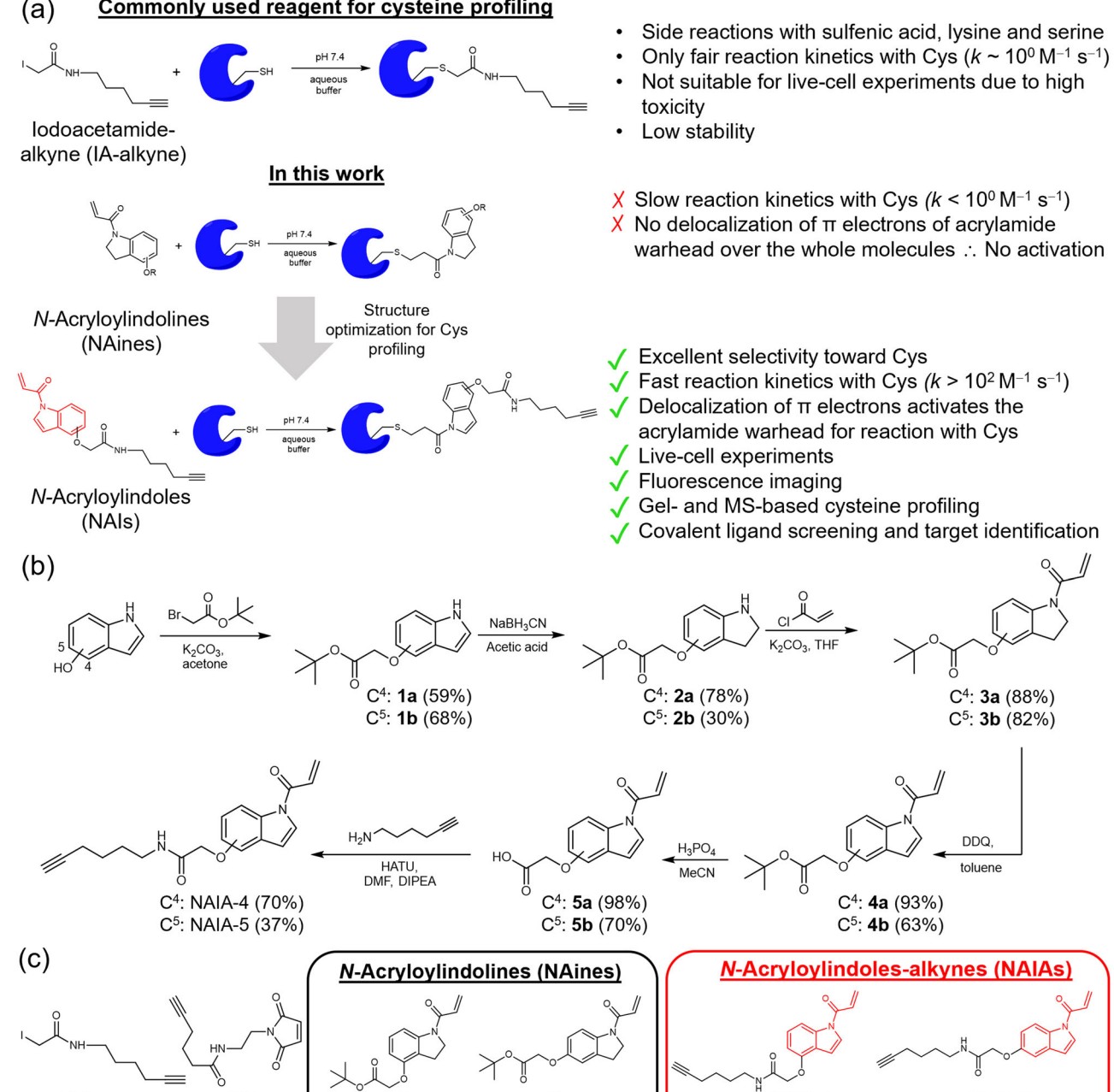

**Fig. 1 | *N*-Acryloylindole (NAI) as a new chemical tool for Cys profiling and imaging. a** Illustration of the attractive features of NAI. Delocalization of π electrons over NAI increases electrophilicity of the acrylamide warhead, resulting in activation of the acrylamide for fast and complete reaction with nucleophilic Cys, which is not found in other acrylamide compounds such as the negative control *N*-acryloylindoline (NAine). Together with good selectivity toward Cys, low cellular

toxicity, and formation of stable Cys-adduct readily detectable by MS, NAI enables both in vitro and live-cell Cys profiling by gel-based and MS-based chemoproteomic experiments. **b** Synthetic scheme for *N*-acryloylindole-alkynes, NAIA-4, and NAIA-5, and negative control compounds, *N*-acryloylindoline **3a** and **3b**. **c** Chemical structures of commonly used cysteine-reactive probes, iodoacetamide-alkyne (IAA) and maleimide-alkyne (MIA), and NAines and NAIAs reported in this study.

capable of labeling cysteines more effectively than IAA in both cell lysates and live cells, where the latter can be explained by fast cellular uptake and fast cysteine reaction kinetics of NAIA in live cells as shown in confocal fluorescence microscopy experiments. Such a fast reaction with cysteines in live cells allows NAIA to image oxidized thiols in cells under oxidative stress by confocal fluorescence microscopy, which is not feasible by IAA. More interestingly, NAIA has been utilized as a probe for MS-based ABPP experiments, both in cell lysates and live cells, and successfully captures larger and unique populations of cysteines than IAA. Many of these cysteines are found to be ligandable by scout ligand CL-Sc in the competitive MS-based ABPP experiment and are potential hotspots to modulate the activities of known cancer drivers. Further application of NAIA for covalent ligand screening allows the discovery of hit compounds targeting new ligandable cysteines and proteins. One of the hit compounds, CL1, has been found to arrest the cell cycle in HepG2 cells at the $G_1$ phase, primarily through binding onto Cys178 of Rac1 and hence inhibition of Rac1 signaling. In view of all these attractive features, NAIA is anticipated to expand the pool of ligandable hotspots in whole-proteome cysteine profiling experiments and to facilitate research on the development of lead compounds for targeting new protein targets for therapy. Its ability to image and profile oxidized cysteines should also help to provide new insights into cysteine redox biology.

## Results

### Design, synthesis, and characterization of NAIs and NAines

In view of the low nucleophilicity of indole nitrogen (even lower than C3 of indole), the key steps for synthesizing NAI (Fig. 1b) involve first reduction of indole to indoline by NaBH$_3$CN in acetic acid, followed by reaction of nucleophilic indoline nitrogen with acryloyl chloride to install the acrylamide warhead and form N-acryloylindoline (NAine). Proceeding with 2,3-dichloro-5,6-dicyanobenzoquinone (DDQ)-mediated oxidation yielded acrylamide-containing indole, NAI, as the final product. To functionalize NAI into a probe, an alkyne handle was introduced to allow click reaction for conjugation to a fluorophore or desthiobiotin[44–46] for ABPP, imaging, and other profiling experiments.

### Cysteine reactivity and selectivity of NAIA

The reactivity of NAIA with cysteine in an aqueous buffer solution has been investigated by LC−MS experiments (Fig. 2). NAIA-5 reacted quickly and completely with N-acetylcysteine methyl ester within 120 s, with the formation of cysteine-adduct as indicated by the new peak at $t = 5.60$ min in the LC (Fig. 2b, c). NAIA-4 was found to react even faster with N-acetylcysteine methyl ester (Supplementary Fig. 1). Whereas the conventional cysteine-reactive probe, IAA, reacted much slower with cysteine than our NAIAs and more than 40% of IAA was left unreacted in the LC−MS experiment (Fig. 2c). Interestingly, NAine compound 3b, which is the reduced form of NAI and lacks the delocalization of π electrons to activate the acrylamide warhead, showed even a slower cysteine reaction than IAA (Fig. 2c), and its biomolecular reaction rate constant was more than 330-fold lower than that of NAIA-5 (Fig. 2d). This highlights the important contribution of aromaticity of indole scaffold to increase electrophilicity of acrylamide warhead for fast and complete cysteine reaction in aqueous buffer solution. It is noteworthy that NAIA-4 and NAIA-5 showed even faster and more complete reactions with cysteine than those recently reported cysteine-reactive compounds with improved cysteine reactivity, such as methylsulfonylbenzothiazole[35,38] (Supplementary Fig. 2) and N-hydroxybenzimidoyl chloride[40] (Supplementary Fig. 3). Maleimide-alkyne (MIA) was found to react faster than NAIAs (Fig. 2c, d). Yet, MIA also showed undesired reactivity with glutathione (GSH), the most abundant cellular thiol, whereas NAIAs showed significantly slower reactions with GSH (Supplementary Figs. 4 and 5). NAIAs also exhibit excellent selectivity toward cysteine over other amino acids (Fig. 2e and Supplementary Fig. 1), typical for acrylamide compounds owing to

the high specificity of thiol-Michael addition reaction, while IAA showed undesired reactivity with serine and lysine (Supplementary Fig. 1c). In view of the susceptibility of hydrolysis of N-acylindole in previous studies[47], we also investigated stability of NAIA-5 in aqueous buffer solution by LC-MS, and found that in PBS solution at pH 7.4, NAIA-5 was stable for at least 3 h (Fig. 2f).

### Cysteine labeling in cell lysates and live cells by NAIA

With the in vitro characterization data in hand, we next moved to investigate the ability of NAIA in labeling functional cysteines in cell lysates and live cells. In the gel-based ABPP experiments, proteins in HepG2 cell lysates labeled by the probe would allow further conjugation to azide-fluor 545 through copper(I)-catalyzed alkyne-azide cycloaddition (CuAAC) and hence showed strong fluorescence. The fluorescence intensity from these labeled protein bands can then reflect the degree of cysteine labeling by the probes. NAIA-5 showed more cysteine labeling than IAA and MIA at all three doses tested with statistical significance (Fig. 3 and Supplementary Fig. 6). Time-dependent experiments revealed quick and more labeling of HepG2 cell lysates by NAIA-5 even at 15 min (Fig. 3b and Supplementary Fig. 7). In addition, NAIA also exhibited excellent performance on labeling reactive cysteines in 231MFP cell lysates, as indicated by higher in-gel fluorescence intensity from NAIA-4 than IAA at all the doses tested (Supplementary Figs. 8a and 9).

In view of the success in labeling cysteines in cell lysates, we then proceeded to apply NAIA to capture reactive cysteines in live cells. The in-gel fluorescence intensities from NAIA-5-treated cells surpassed those from IAA-treated cells (Fig. 3d, g, and Supplementary Fig. 10), particularly at 15 min incubation (>3-fold difference at 10 μM). Similar results of stronger in-gel fluorescence were also found in 231MFP cells incubated with NAIA-4, as compared to the IAA-treated cells (Supplementary Figs. 8b and 11), showing that the superior performance of cysteine labeling by NAIAs is not limited to a particular cell line. The better cysteine labeling can be attributable to the fast reaction kinetics of NAIAs with cysteines, as supported by LC-MS experiments (Fig. 2), and the fast cellular uptake of the probe by HepG2 cells which we will show later in the confocal fluorescence imaging experiment (Fig. 4).

To investigate the underlying mechanism of better cysteine labeling in live cells by NAIA, we conducted confocal fluorescence imaging experiments of HepG2 cells treated with NAIA-5 and IAA respectively. Significantly higher fluorescence intensities were found in cells treated with NAIA-5 than IAA for all the three-time points tested (15, 30, and 60 min; Fig. 4a, b), indicating more labeling of reactive cysteines by NAIA-5. Notably, cells treated with NAIA-5 for 15 min already showed very strong fluorescence intensity, while IAA-treated cells showed much weaker fluorescence at 15 min (Fig. 4a, b). This suggests that NAIA-5 showed good cellular permeability and fast reaction kinetics with cysteines in live cells, where the latter is consistent with our in vitro findings from LC−MS experiments (Fig. 2), thus allowing NAIA-5 to quickly capture and label reactive cysteines in cells. NAIA-5 shows good labeling of proteins in both cytosol and nucleus, while the labeling of nuclear proteins by IAA is relatively lower, as indicated by the smaller Manders' colocalization coefficient (Supplementary Table 1). HepG2 cells with elevated GSH levels did not show significant changes in fluorescence, suggesting minimal effects of GSH on NAIAs in labeling functional cysteines (Supplementary Fig. 12). In addition, NAIA-4 and NAIA-5 exhibited much lower cytotoxicity than IAA as shown in WST-8 and MTT assay respectively (IC$_{50}$ > 40 μM; Fig. 4c and Supplementary Fig. 13), which is another attractive feature for live cell experiments.

### Confocal fluorescence microscopy imaging of thiol oxidation

Global thiol trapping techniques such as OxICAT (ICAT = Isotope-coded Affinity Tag)[48–50] have demonstrated successes in studying thiol oxidations, but the highly dynamic nature of thiol oxidative

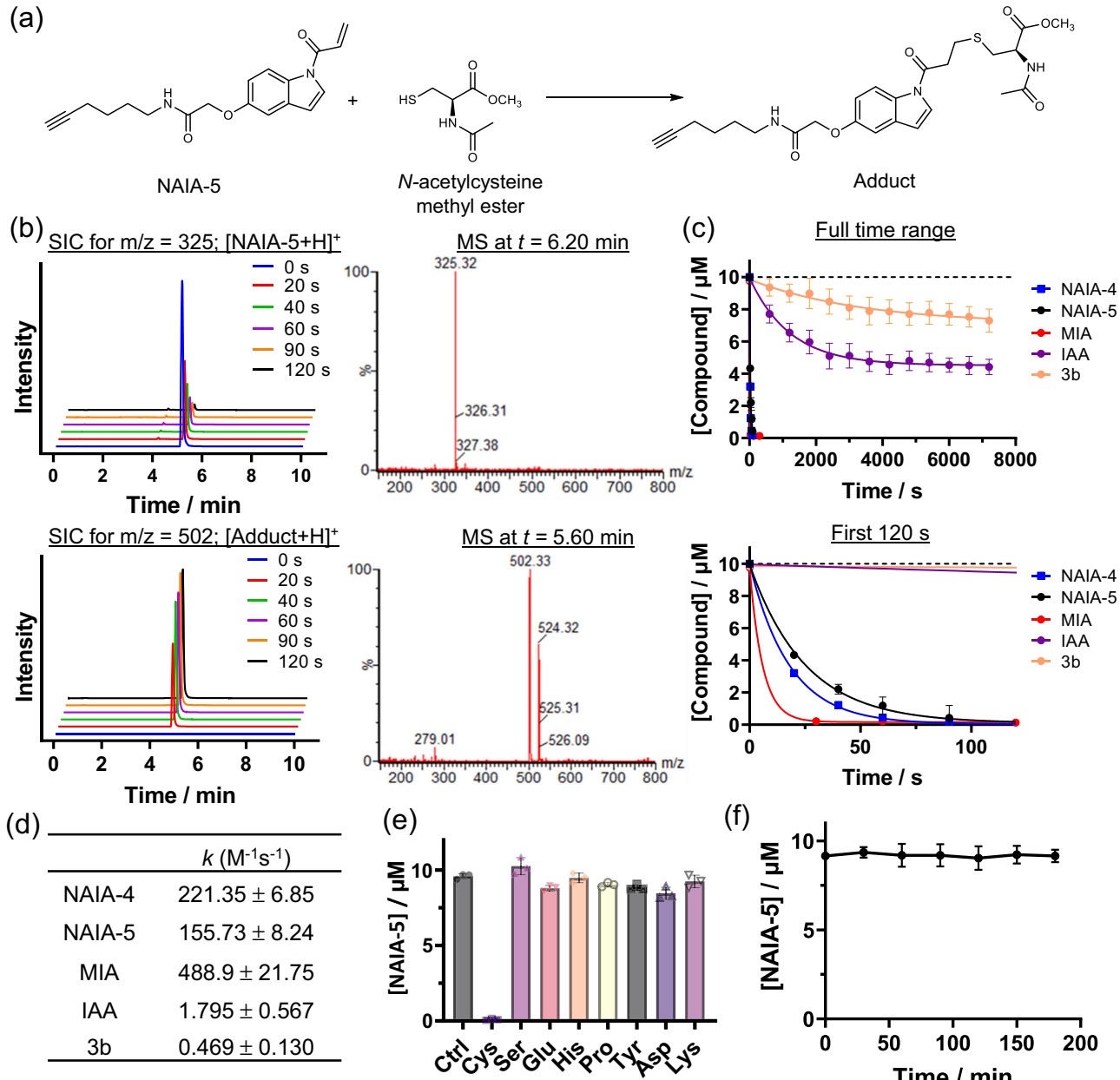

**Fig. 2 | LC−MS analysis on the reactions between Cys-reactive compounds and small amino acids in aqueous buffer solution. a** Chemical equation showing the thiol-Michael addition of NAIA-5 with *N*-acetylcysteine methyl ester (*N*-Ac-Cys-OMe), a N- and C-terminal protected cysteine, to form the adduct product. **b** NAIA-5 (10 μM) in aqueous buffer solution (PBS-MeOH, 4:1, v/v) was incubated with *N*-Ac-Cys-OMe (250 μM). At indicated time intervals, an aliquot of the solution mixture was sent for LC−MS analysis. Selected ion chromatograms (SIC) at *m/z* = 325 and 502, corresponding to the molecular ion of [NAIA-5 + H]⁺ and [Adduct+H]⁺ respectively. The mass spectra (MS) at 6.20 and 5.60 min confirm the identity of NAIA-5 and adduct. **c** Changes in Cys-reactive compound concentration over time upon incubation with *N*-Ac-Cys-OMe (250 μM). **d** Biomolecular rate constant of reactions between cysteine-reactive compounds and *N*-Ac-Cys-OMe. **e** Changes in the level of NAIA-5 after incubation with amino acids (30 μM) for 30 min (*n* = 3). **f** Stability of NAIA-5 in the aqueous buffer solution in the absence of Cys (*n* = 3). Quantified data were shown on average ± SD from *n* = 3 different replicates/groups.

modifications may result in loss of these modifications throughout the cell lysis process. Trapping free thiols in live cells could be a good strategy to prevent the loss of oxidative modifications, but this has not been realized. In view of the fast and better labeling of reactive cysteines in live cells, as well as the compatibility of NAIAs with confocal fluorescence imaging, we applied NAIAs as imaging probes for studying dynamic changes of thiol reactivity/modifications in cells facing oxidative stress. We first sought to capture and image free thiols by NAIA-5 in HepG2 cells by confocal fluorescence microscopy (Fig. 5a). For the cells pretreated with solvent control only, we observed strong fluorescence from NAIA-5 labeling, while almost no

fluorescence can be detected in IAA-labeled cells using the same imaging parameters (Fig. 5b). This suggests much faster labeling and trapping of free thiols by NAIA-5 than IAA. A $H_2O_2$ dose-dependent decrease in cellular fluorescence intensity was found in cells labeled by NAIA-5, and restored when co-incubated with anti-oxidant *N*-acetylcysteine (NAC) (Fig. 5c), illustrating that NAIA-5 is capable of probing changes in cellular thiol oxidation. Interestingly, we cannot observe statistically significant changes in fluorescence intensity from IAA-labeled cells upon $H_2O_2$ stimulation (Fig. 5d), even at high laser power to better capture the weaker fluorescence from IAA-labeled cells.

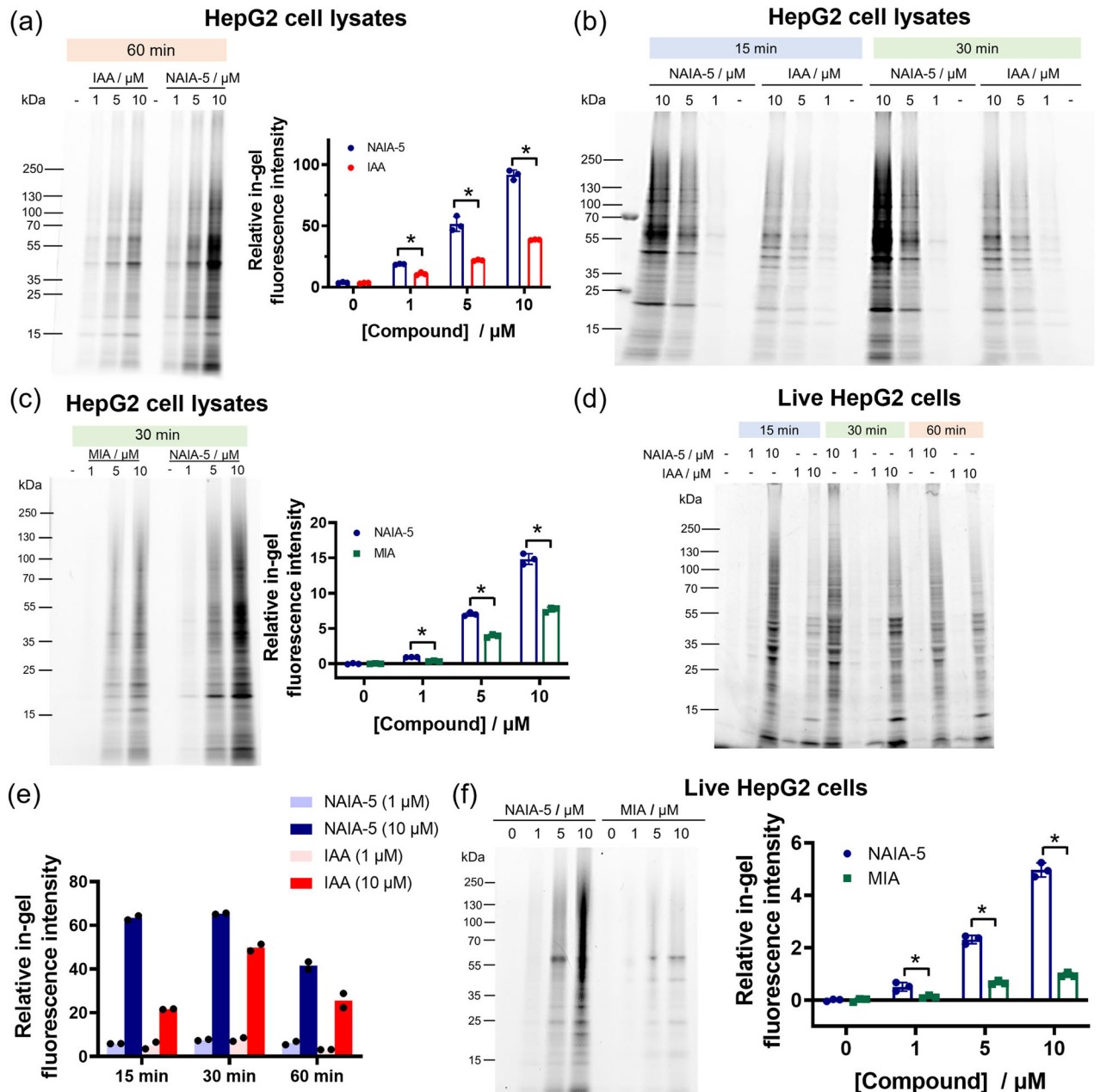

**Fig. 3 | Gel-based ABPP experiments demonstrate excellent Cys labeling of cell lysates and live cells by NAIA-5. a, b** Gel-based ABPP analysis of NAIA-5 and IAA on Cys labeling of HepG2 cell lysates in vitro. HepG2 cell lysates (100 μg) were incubated with NAIA-5 or IAA at indicated concentrations and time intervals, followed by CuAAC reaction with azide-fluor 545 (25 μM). The labeled proteins were then boiled with sampling buffer, and read out by in-gel fluorescence after SDS-PAGE. **c** Gel-based ABPP analysis of NAIA-5 and MIA on Cys labeling of HepG2 cell lysates in vitro. **d** HepG2 cells were incubated with NAIA-5 or IAA in a complete medium at indicated concentrations and time intervals. The cells were then lysed, labeled with azide-fluor 545 (5 μM) by CuAAC reaction, boiled with sampling buffer, and the labeled proteins were visualized by in-gel fluorescence after SDS-PAGE. **e** Quantification data for the experiments in (**d**). **f** In-gel fluorescence from live HepG2 cells treated with NAIA-5 or MIA in complete medium for 30 min at indicated concentrations. Quantified data were shown on average ± SD from $n = 3$ replicates/group, except $n = 2$ replicates/group in (**e**). Statistical analyses were performed with unpaired two-tailed Student's $t$-tests. Statistical significance is expressed as *$P = 4.18 \times 10^{-4}$, $1.07 \times 10^{-3}$, and $2.01 \times 10^{-5}$ for 1, 5, and 10 μM, respectively, in (**a**); *$P = 3.22 \times 10^{-5}$, $5.19 \times 10^{-5}$ and $7.36 \times 10^{-5}$ for 1, 5, and 10 μM, respectively, in (**c**); *$P = 1.51 \times 10^{-2}$, $7.42 \times 10^{-5}$, and $1.54 \times 10^{-5}$ for 1, 5, and 10 μM, respectively, in (**f**).

This would be more advantageous to label and image oxidized thiols instead of unoxidized ones for better determination of their identity and subcellular localization. Therefore, we utilized the NAI/NAIA couple to capture oxidized thiols and image them by confocal fluorescence microscopy (Fig. 5e). A $H_2O_2$ dose-dependent increase in fluorescence intensity was found (Fig. 5f), indicating the success of our NAI/NAIA couple to capture and image oxidized thiols as well as monitoring their elevated levels in cells facing oxidative stress.

NAC co-incubation with $H_2O_2$ was found to attenuate the enhanced fluorescence intensity (Fig. 5f). In view of the fact that many reported thiol trapping techniques are limited to cell lysates[8,48,49] and not applicable for imaging experiments, the readiness of NAI/NAIA to trap/label cellular thiols in live cells enables them to function as imaging probes to investigate oxidative modifications and other dynamic changes in cellular thiol activity with high spatial resolution.

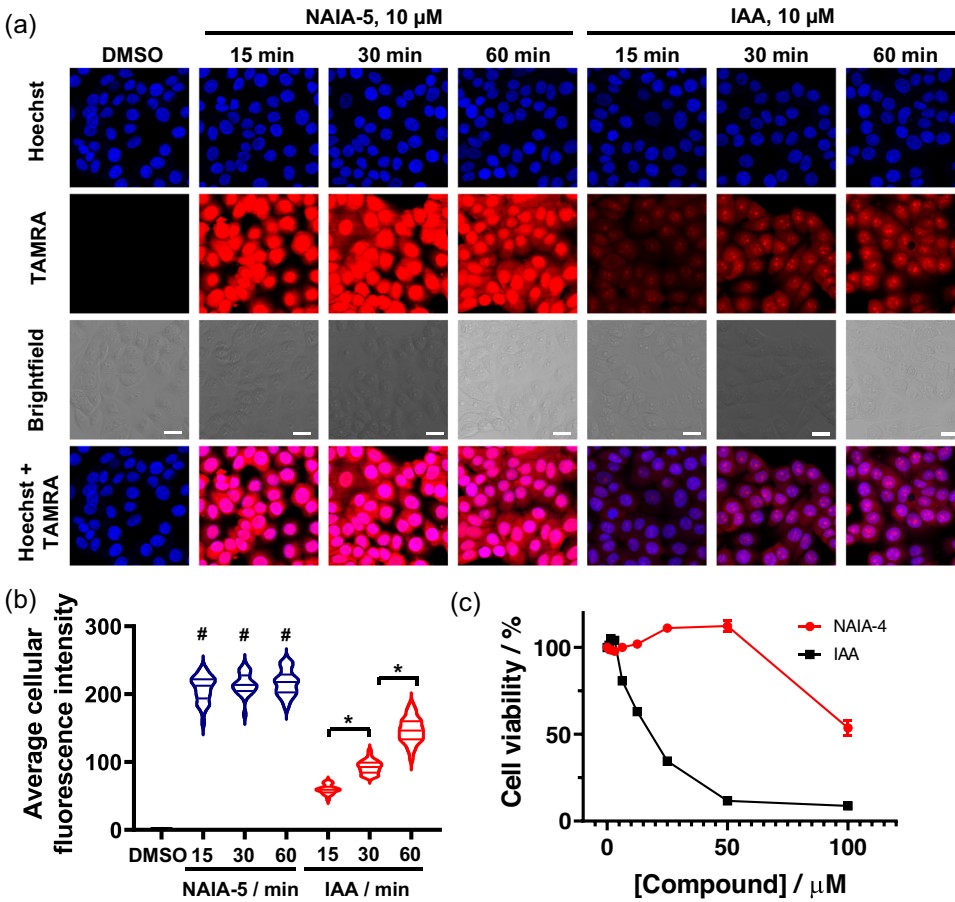

**Fig. 4 | Confocal fluorescence imaging and cell viability assay reveal the superior performance of NAIA on thiol labeling in live cells. a** HepG2 cells on the 8-well chambered slide were incubated with NAIA-5 or IAA (10 μM) in complete medium for the indicated time intervals. The cells were then washed with PBS, fixed by 4% paraformaldehyde, permeabilized, and reacted with azide-fluor 545 (20 μM) by CuAAC reaction at room temperature in the dark for 1 h. The stained cells were washed with PBS and further incubated with Hoechst (8.2 μM) for 15 min. The cells were then washed with PBS and imaged by confocal fluorescence microscopy. **b** Cellular fluorescence intensity as determined by ImageJ. Quantified data were

from $n = 30$ cells from 3 different biological replicates/groups, with median, first, and third quartile showed. Statistical analyses were performed with unpaired two-tailed Student's $t$-tests. Statistical significance is expressed as $^{\#}P = 7.52 \times 10^{-53}$, $1.99 \times 10^{-59}$, and $3.84 \times 10^{-55}$ for incubation with NAIA-5 for 15, 30, and 60 min, respectively, as compared to control; $^{*}P = 8.03 \times 10^{-22}$ and $1.95 \times 10^{-20}$ for incubation with IAA for 15 vs. 30 min, and 30 vs. 60 min, respectively. **c** Cell viability of 231MFP cells after incubation with NAIA-4 or IAA for 2 h, as indicated by WST-8 assay. Quantified data were shown in average ± SD from $n = 6$ different biological replicates/groups. Scale bar = 20 μm.

## MS experiments to identify oxidized cysteines by NAI/NAIA-5

The success in imaging thiol oxidations by NAI/NAIA prompted us to further investigate their ability in the identification of oxidized cysteines. Through click chemistry to install desthiobiotin instead of rhodamine used in the imaging experiments, this enables us to enrich NAIA-5-labeled peptides and analyze them by MS experiment (Fig. 6a). ~30,000 NAIA-5-labeled peptides on average were detected in HepG2 cells treated with solvent vehicle, $H_2O_2$ and $H_2O_2 + NAC$ (184,644 modified peptides in the aggregate; Fig. 6b and Supplementary Data 1), with a total of 12,584 cysteines being identified. Among these cysteines, 5625 cysteines have not been reported by the powerful cysteine-reactive phosphate tag (CPT) platform for redox proteomics[51] (Supplementary Fig. 14), highlighting the robust application of NAIA-5 as a cysteine probe for MS experiment. By comparing the LFQ intensity of cysteines from control samples and that from cells treated with $H_2O_2$ (Fig. 6c), this allows us to examine any oxidative modifications on the identified cysteines. 583 cysteines in $H_2O_2$-stimulated cells showed higher LFQ intensity (>4-fold as compared to that in control), indicating that they were highly oxidized by $H_2O_2$. Whereas there were 112 cysteines with lower LFQ intensity (<0.25-fold) in the $H_2O_2$-stimulated cells. Gene Ontology (GO) analysis[52] revealed that these cysteines associated more closely with responses to hydrogen peroxide and

oxidative stress as compared to other identified cysteines (Fig. 6d). This suggests changes in reactivity of these cysteines, probably through reduction or irreversible oxidation, after $H_2O_2$ stimulation.

We have examined the LFQ intensity of cysteines on important classes of proteins in response to oxidative stress (Fig. 6e). Most of the identified cysteines in the PRDX protein family showed a lower degree of oxidation in cells stimulated with $H_2O_2$, e.g., C52 on PRDX1 which is the active site. This can be rationalized by the active reduction of oxidized disulfide bonds on PRDX through oxidoreductase enzymes for detoxifying $H_2O_2$, consistent with those reported in literature[53]. The only exception is C83 on PRDX1, suggesting its different roles in the cellular response to $H_2O_2$. For thioredoxin and related proteins, C54 on TXNRD2, which is within the binding site of cofactor FAD, was found to be the most sensitive cysteine toward oxidation by $H_2O_2$. We also found oxidative modification on C102 and C134 on GPX4 by $H_2O_2$, with C102 showing a greater extent of oxidation. On the other hand, C247 on GAPDH, which is the site for nitrosylation and succinylation and is important for glycolytic activity, revealed a lower cysteine reactivity in $H_2O_2$-treated cells. This indicates a link between oxidative stress and metabolic processes. For proteins associated with glutathione metabolism, they showed very different responses, with C150 on GSTCD, C87 on GSTM1 and C190 on GSTM3 being highly oxidized, while the

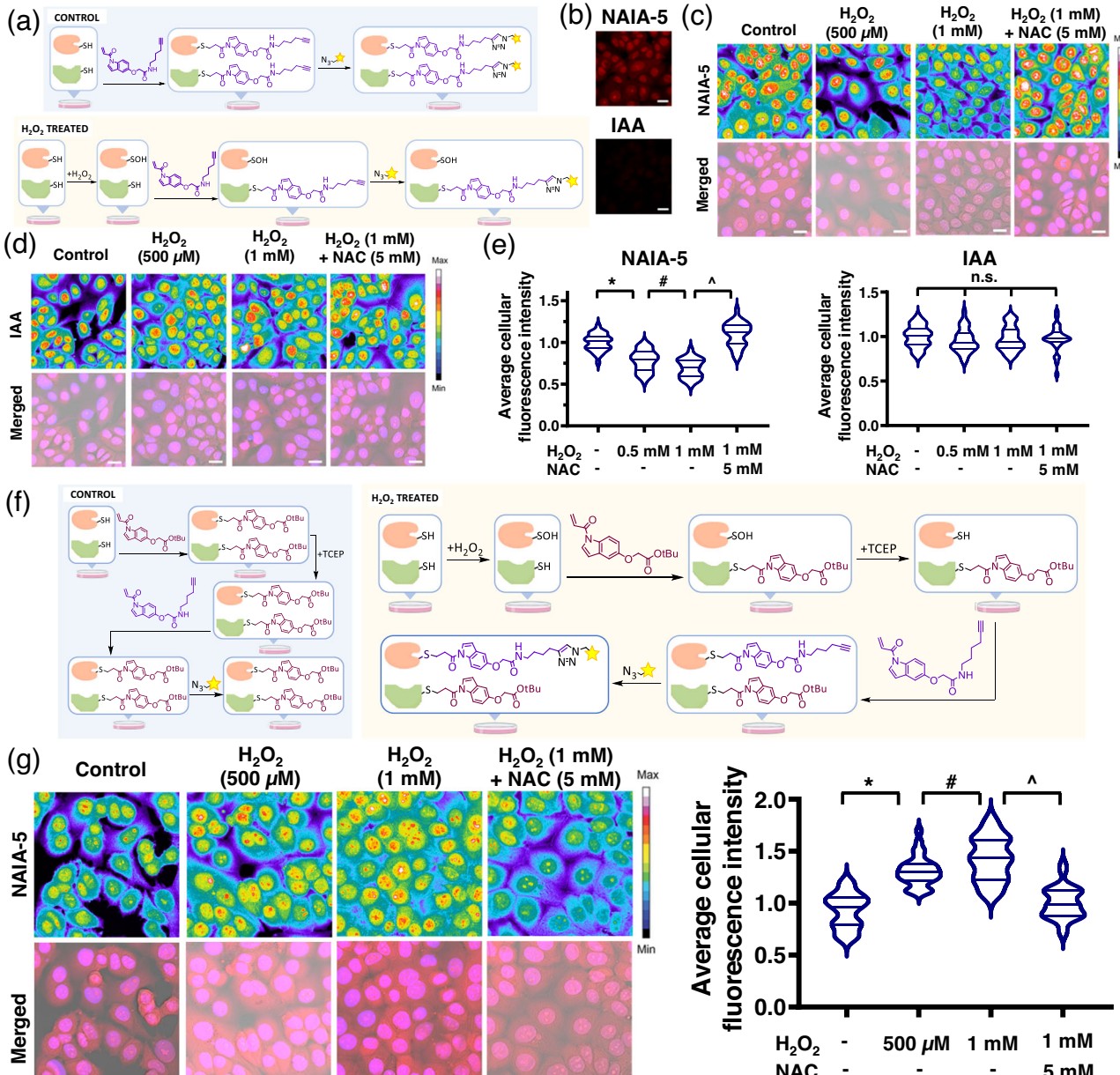

**Fig. 5 | NAIA-5 allows successful imaging of changes in thiol reactivity in cells under oxidative stress by confocal fluorescence microscopy. a** Schematic cartoon illustrating the experimental setup to monitor levels of reduced cellular thiols by NAIA-5. **b** Confocal fluorescence microscopy images of HepG2 cells after incubation with NAIA-5 and IAA (50 μM, 10 min). The two images were recorded by the same imaging parameters. Similar results were found in three different biological replicates/groups. **c, d** Confocal fluorescence microscopy images of HepG2 cells probed by NAIA-5 or IAA. **e** Quantification data in (**c**) and (**d**). **f** Schematic cartoon illustrating the working principle of NAI/NAIA-5 couple to label and image oxidized thiols. HepG2 cells were first pretreated with solvent vehicle, H$_2$O$_2$ (0.5 or 1 mM), or a mixture of H$_2$O$_2$ and N-acetylcysteine (NAC; 5 mM) in a complete medium for 15 min. The cells were then washed with PBS and incubated with a high concentration of NAI compound **3** (50 μM) in a complete medium at 37 °C for 10 min to react with the free thiols. Then, the cells were washed with PBS, fixed with 4% paraformaldehyde, and permeabilized by PBS with 0.5% Triton X-100. The fixed cells were treated with TCEP (1 mM) in PBS for 1 h to reduce oxidized thiols, followed by incubation with NAIA-5 (10 μM) in PBS to label the newly formed free thiols. The cells were washed with PBS, followed by CuAAC reactions with azide-fluor 545 (4 μM). After the reaction, the cells were washed and imaged in PBS by a confocal fluorescence microscope. **g** Confocal fluorescence microscopy images of HepG2 cells probed by NAI/NAIA-5 couple. Merged images are composed of images from brightfield, NAIA-5, and Hoechst channels. Cellular fluorescence intensity was determined by ImageJ. Quantified data were from $n = 30$ cells from three different biological replicates/groups, with median, first, and third quartile showed. Statistical analyses were performed with unpaired two-tailed Student's $t$-tests. Statistical significance is expressed as *$P = 8.00 \times 10^{-10}$, #$P = 5.88 \times 10^{-3}$ and ^$P = 1.66 \times 10^{-17}$ respectively in (**e**); *$P = 2.70 \times 10^{-13}$, #$P = 1.66 \times 10^{-2}$ and ^$P = 2.15 \times 10^{-12}$ respectively in (**g**); n.s. denotes not significant. Scale bar = 20 μm. Illustrations in **a** and **f** were created using ChemDraw, BioRender.com, and MS Powerpoint.

reactivity of C26 on GSTT1 decreased significantly in the presence of H$_2$O$_2$ and the antioxidant NAC, suggesting a reduction or irreversibly modification. Proteins from mitochondrial complex 1 also revealed very different reactivities, with C332 on NDUFV1 as a notable example, showing enhanced oxidation by H$_2$O$_2$ but low reactivity upon the addition of NAC (Fig. 6e). This illustrates its specific roles in response to dynamic changes of cellular redox state. All the above findings illustrate the highly complex redox biology of cysteines, and our NAI/NAIA-5 couple is successful in identifying these important cysteines and their oxidations.

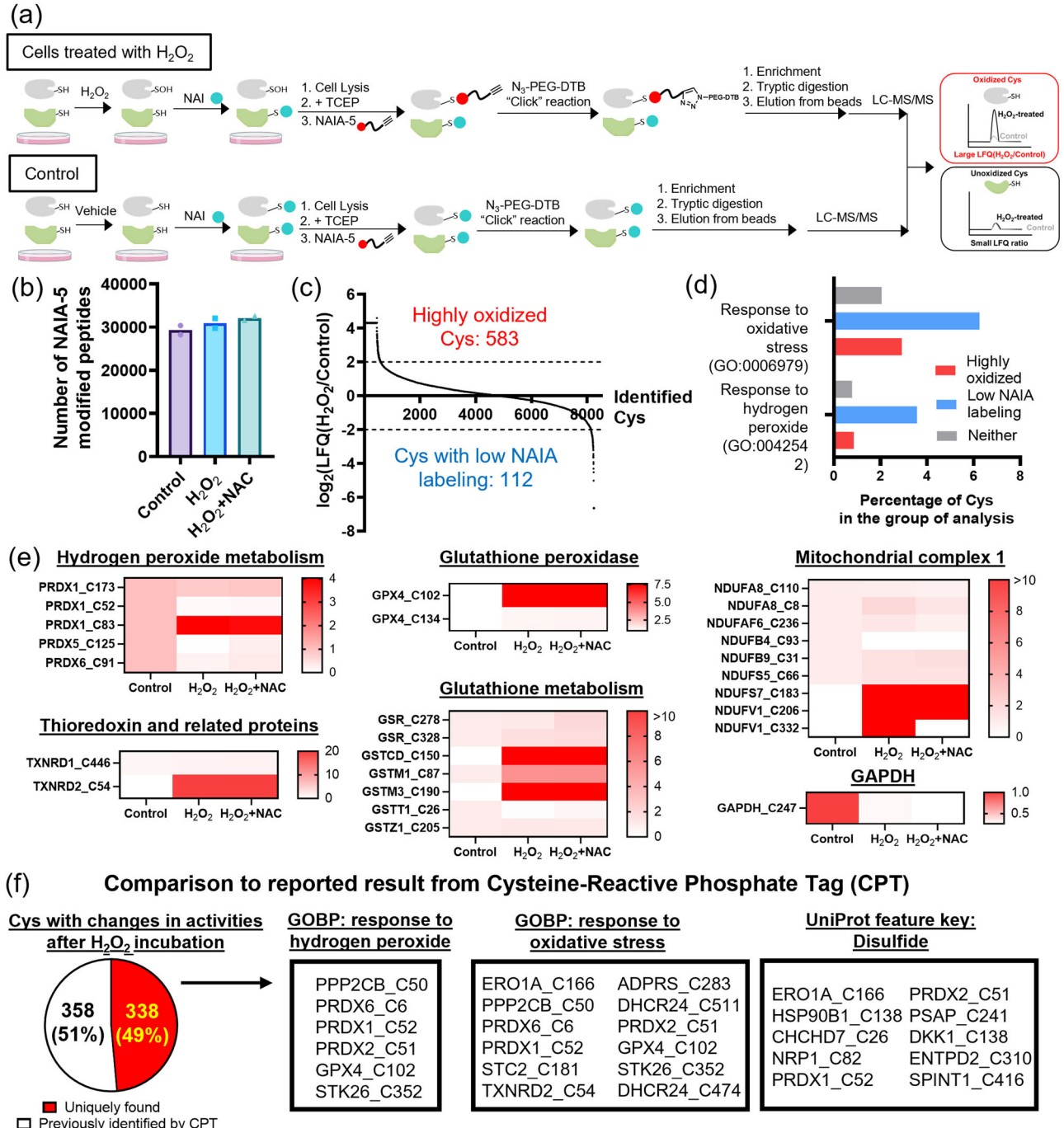

**Fig. 6 | NAI/NAIA-5 couple allows identification of oxidized cysteines in cells under oxidative stress by MS experiment. a** Schematic cartoon illustrating the working principle of NAI/NAIA-5 couple in MS to identify oxidized cysteines, which show a larger LFQ ratio when comparing $H_2O_2$-treated cells with control. Created using ChemDraw, BioRender.com, and MS Powerpoint. **b** Number of NAIA-5-modified peptides detected in the MS experiments. Quantified data were the average values from $n = 2$ different replicates/groups. **c** Analysis of the LFQ intensity of NAIA-5 labeled peptides in $H_2O_2$-stimulated cells vs control revealed 583 highly oxidized cysteines in cells pretreated with $H_2O_2$, while 112 cysteines showed low NAIA-5 labeling in the $H_2O_2$-stimulated cells. **d** GO analysis on the percentage of the identified cysteines in response to hydrogen peroxide and oxidative stress. **e** Heatmap showing LFQ intensities (normalized by the intensity of the control group) of important classes of cysteines in response to oxidative stress. **f** Comparison of cysteines with significant changes in reactivity identified by our NAI/NAIA-5 with the results from reported Cysteine-reactive Phosphate Tag (CPT). We have successfully identified 338 unique cysteines, including those that are important for cellular responses to hydrogen peroxide and oxidative stress and can undergo disulfide oxidation.

We compared the 696 cysteines with changes in activity under $H_2O_2$ treatment with the reported redox proteomics results from CPT[51]. Our NAI/NAIA-5 couple enables profiling 338 unique redox-sensitive cysteines that were not reported by CPT (Fig. 6f). Importantly, these cysteines are associated closely with cellular response to oxidative stress and are capable of mediating oxidative modifications (Fig. 6f).

This showcases the values of using NAI/NAIA couple to profile these new cysteines for studying thiol oxidation as well as cysteine redox biology.

**MS-based chemoproteomics on profiling functional cysteines**
In addition to studying cysteine redox biology, we are interested in applying NAIAs for MS-based ABPP experiments (Supplementary

Fig. 15)[9,11,12] to identify functional cysteines in cancer cells which can be hotspots for therapy. We picked NAIA-5 as the cysteine probe in this first set of MS experiments, instead of NAIA-4, because of the higher stability of NAIA-5-Cys adduct as shown in the LC−MS experiment (with no observable degradation over 48 h; Supplementary Fig. 16). This could be important in view of the long preparation time for this MS experiment (Supplementary Fig. 15). It is noteworthy that in this experiment, the NAIA-5-treated sample was first mixed with IAA-treated sample before tryptic digestion and LC-MS/MS analysis, thus any difference found in labeling by NAIA-5 or IAA should not be attributable to the discrepancy in sample preparation or MS running condition and this allows good comparison of NAIA-5 with IAA for proteome-wide cysteine profiling. Also, the incubation concentration of IAA with the cell lysates (10 μM) was much lower than its working concentration reported in previous studies for cysteine profiling (>100 μM)[3,5,9,11,12,50], as we are aiming to develop a better cysteine probe that can profile cysteine effectively even at low working concentrations and NAIA-5 has been found to label cysteines well at 10 μM in the gel-based ABPP experiments (Fig. 3). A short 2 h-gradient LC run was used in this experiment to examine the ability of NAIA-5 to profile cysteine by this simple setup.

A significantly larger number of peptides modified by NAIA-5 than IAA were found in all the triplicate samples (average numbers with modifications by NAIA-5 and IAA are 5269 and 957, respectively; Fig. 7a and Supplementary Data 2). Much higher signal intensities were found from the experiment using NAIA-5, as compared to IAA (>5.5-fold; Fig. 7e), indicating the good ionization of NAIA-5-modified peptides for MS experiments. NAIA-5 also profiled a larger population of cysteines than IAA in aggregate (6387 vs. 1257; Fig. 7b), with less undesired off-target labeling onto other amino acids than IAA as determined by unbiased open search by MSFragger[54,55] (Supplementary Fig. 17). These profiled cysteines by NAIA-5 were on 3394 different proteins, showing overlap with only 905 proteins which were also labeled by IAA, while the remaining 2489 proteins represent the unique pool of proteins with functional cysteines profiled by NAIA-5 in this experiment (Fig. 7c). When compared to the study with the largest pool of functional cysteines identified by an IAA derivative, DBIA[56], NAIA-5 is still capable of expanding the pool significantly by uniquely profiling 530 proteins, and more importantly 2990 functional cysteines (>46% of the total number of Cys probed by NAIA-5 in the experiment; Fig. 7d and Supplementary Data 3). Comparison to other reported cysteine profiling experiments using IAA under optimal conditions[29] or with advanced chemically cleavable linker[30] also revealed identifications of a large population of unique functional cysteines (>2700) by NAIA-5 (Supplementary Fig. 18). It is noteworthy that the working concentration of NAIA-5 (10 μM) in our experiment is far lower than DBIA in the reported study (500 μM), and a further increase in a number of profiled cysteines and proteins by NAIA-5 is anticipated at higher working concentrations. Recently, a cysteine chemoproteomics database has been reported[57], and compared to this large dataset from different chemoproteomics experiments, NAIA-5 can still identify 308 unique cysteines on 260 different proteins from our single profiling experiment with working concentration at 10 μM (Supplementary Data 4). All these results highlight that NAIA-5 is comparable to, if not better than, the optimized and current state-of-the-art cysteine-reactive probes.

To investigate the properties of labeled proteins by NAIA-5 and IAA, we utilized the DrugBank database[58] and GO analysis[52] to examine the availability of pharmacophores to drug these labeled proteins and their biological functions respectively. 84% of proteins labeled by NAIA-5 were not on the list of DrugBank (non-DrugBank proteins) and it is more than the 76% found for IAA (Fig. 7f). More interestingly, for the proteins on the list of DrugBank (DrugBank proteins) with NAIA-5 labeling, they are mostly enzymes according to GO analysis. The non-DrugBank proteins show very different profiles in terms of their biological functions, with proteins involved in gene expression and

regulation as the largest category (Fig. 7f). Further GO analysis on biological processes regulated by the profiled proteins from NAIA-5 revealed 650 processes which are unique for NAIA-5 and not found in IAA profiling, with a significant contribution from processes associated with gene expression and regulation on the top 20 unique processes (Fig. 7g; highlighted in red). In view of the fact that transcription factors and their associated proteins are often considered as one of the most important protein targets for therapy but remain mostly undrugged[11,12,17,28,59,60], this is anticipated that NAIA-5, when it is employed in competitive binding experiments with covalent ligands, can be a valuable tool for identifying new covalent drug lead compounds to target these important proteins. This will be illustrated in the later section.

We also performed in-cell cysteine profiling experiments by treating live HepG2 cells with NAIA-5 and IAA (100 μM) for 30 min, followed by cell lysis and MS sample preparation as shown in Supplementary Fig. 15. We found much more probe-modified peptides from NAIA-5 than IAA in duplicate experiments (Fig. 7h and Supplementary Data 5), as well as higher total ion intensity and a larger population of uniquely profiled cysteines and proteins by NAIA-5 (Supplementary Fig. 19a–c). Lower percentage of Cys labeling on peptides with GCD local sequence was observed in cells treated with NAIA-5 than in IAA (Supplementary Fig. 19d–f). This can explain, at least partly, why NAIA-5 is relatively less reactive with glutathione (a short GCD peptide) as compared to IAA. Interestingly, by comparing the results of MS experiments on HepG2 cell lysates and live cells, a significant amount of cysteines (1384) were profiled in live cells only but not in cell lysates (Fig. 7i), and 57% of them have not been identified by DBIA in the current state-of-the-art profiling experiment[56] which is higher than the percentage of new cysteines identified in both experiments (42%). Among the unique cysteines identified in the live cell experiment only, more proteins were found to have only 1 probe-modified cysteine, as compared to those profiled in both experiments. A larger percentage of modified proteins in live cell experiments only are found to be non-DrugBank proteins, and the distribution of protein class is very different from those profiled in both live cell and cell lysate experiments. More functional cysteines were found on enzymes, scaffolding proteins, modulators and adapters in the live cell experiment (Fig. 7j). All these results suggest the presence of highly reactive cysteines in live cells, and they might lose their reactivity after cell lysis and hence they cannot be identified by MS experiment on cell lysates. Therefore, the superior performance of NAIA in live cell experiments should aid in discovering a unique pool of functional cysteines.

We noted that IAA has been widely used for cysteine profiling of cell lysates by Multidimensional Protein Identification Technology (MudPIT)[2,3], which can significantly enhance peptide separation and hence increase resolution and a number of identified peptides in LC−MS/MS experiments. We have performed the MS experiment with MudPIT, and found that NAIA-4 showed better performance than IAA for cysteine profiling in 231MFP cell lysates (Supplementary Fig. 20a and Supplementary Data 6), similar to the finding of NAIA-5 in the experiment with HepG2 cell lysates (Fig. 7c), supporting the robust applications of NAIAs for proteome-wide cysteine profiling in different biological samples and different experimental setups.

## NAIA-modified cysteines and proteins are ligandable

We employed a cysteine-reactive covalent ligand, CL-Sc, as the scout compound to investigate ligandability of NAIA-modified cysteines and proteins. By competitive MS-based ABPP platform[6], LFQ ratios of peptides (LFQ of the control sample/LFQ of the sample treated with CL-Sc) can estimate cysteine binding by CL-Sc (Fig. 8a). With reference to the previous study of DBIA[56], a threshold of 4 was set in our analysis (75% occupancy of the cysteine site by CL-Sc) and peptides with larger ratios are considered as ligandable. We found 1177 and 1006 cysteines

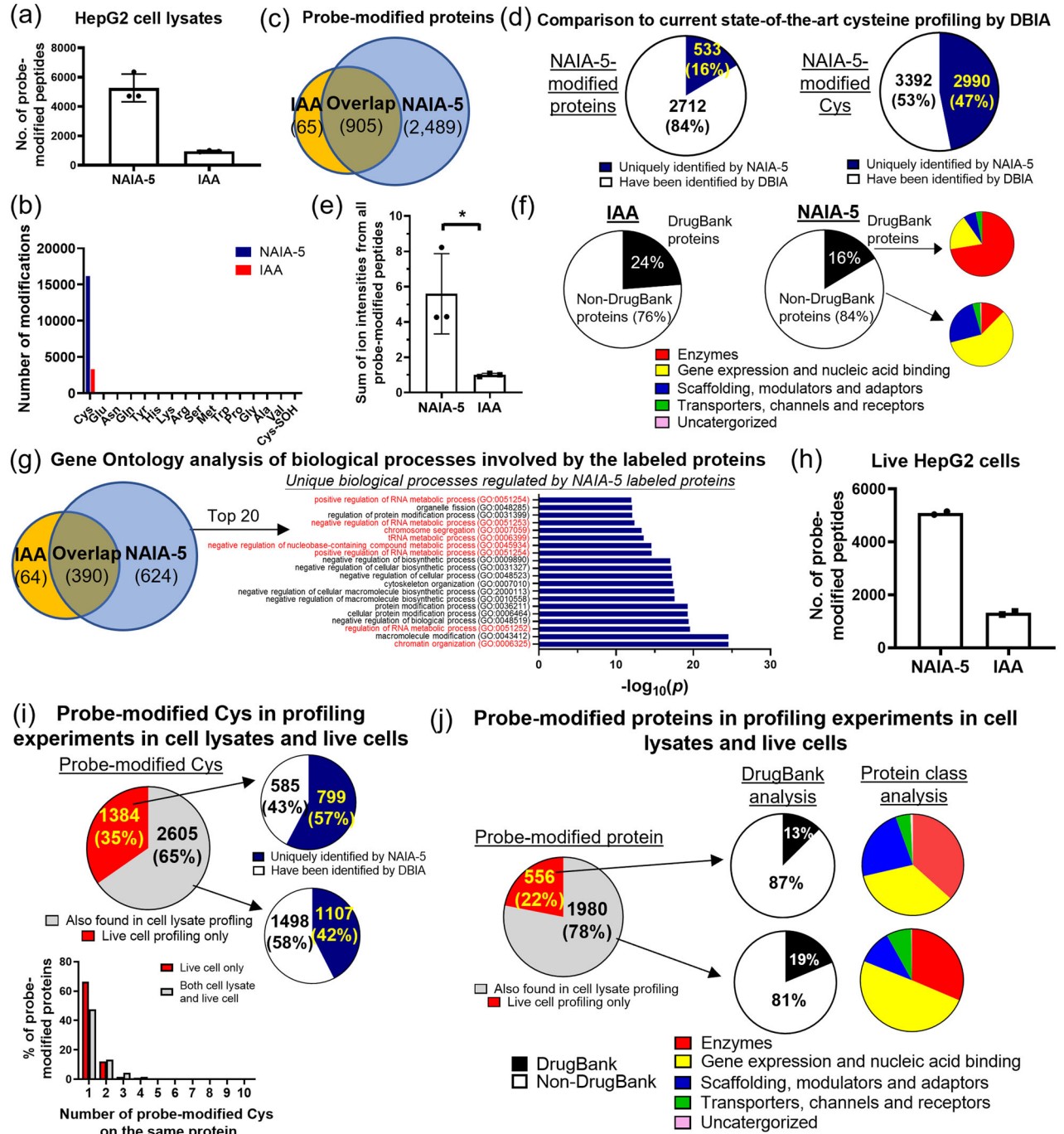

**Fig. 7 | LC–MS/MS-based chemoproteomics experiments to investigate Cys profiling in HepG2 cell lysates and in live HepG2 cells by NAIA-5 and IAA.**
**a** Number of probe-modified peptides in HepG2 cell lysates identified in the MS experiment outlined in Supplementary Fig. 15. Quantified data were shown in average ± SD from $n = 3$ different replicates/group. **b** Modifications on different amino acids by NAIA-5 and IAA, respectively, as identified from the MS experiment. Cys-SOH: sulfenic acid form of Cys. **c** Number of probe-modified proteins identified in the triplicate experiment. **d** Comparison of the proteins and cysteines profiled by NAIA-5 with those profiled by an IAA-derived probe, DBIA, in the current state-of-the-art paper for cysteine profiling[56]. The full list of profiled proteins and cysteines by NAIA-5 can be found in Supplementary Data 2. **e** Relative intensity of the sum of signals from probe-modified peptides by NAIA-5 or IAA. Quantified data were

shown on average ± SD from $n = 3$ different replicates/groups. Statistical analyses were performed with unpaired two-tailed Student's $t$-tests. *$P = 0.023$. **f** Analysis of probe-modified proteins by NAIA-5 and IAA, respectively, using the DrugBank database, and the functions of these proteins by Gene Ontology database. **g** Gene Ontology analysis of the biological processes involving the probe-modified proteins, with the top 20 unique processes regulated by NAIA-5-modified proteins shown in the bar chart. $P$ values were determined by Fisher's exact test. Those highlighted in red are associated with gene expression and regulation. **h** Number of probe-modified peptides identified by incubation of live HepG2 cells with NAIA-5 or IAA. Quantified data were the average values from $n = 2$ different replicates/groups. **i** Comparison of probe-modified cysteines and **j** probe-modified proteins identified by NAIA-5 in cell lysates with those in live cell experiment.

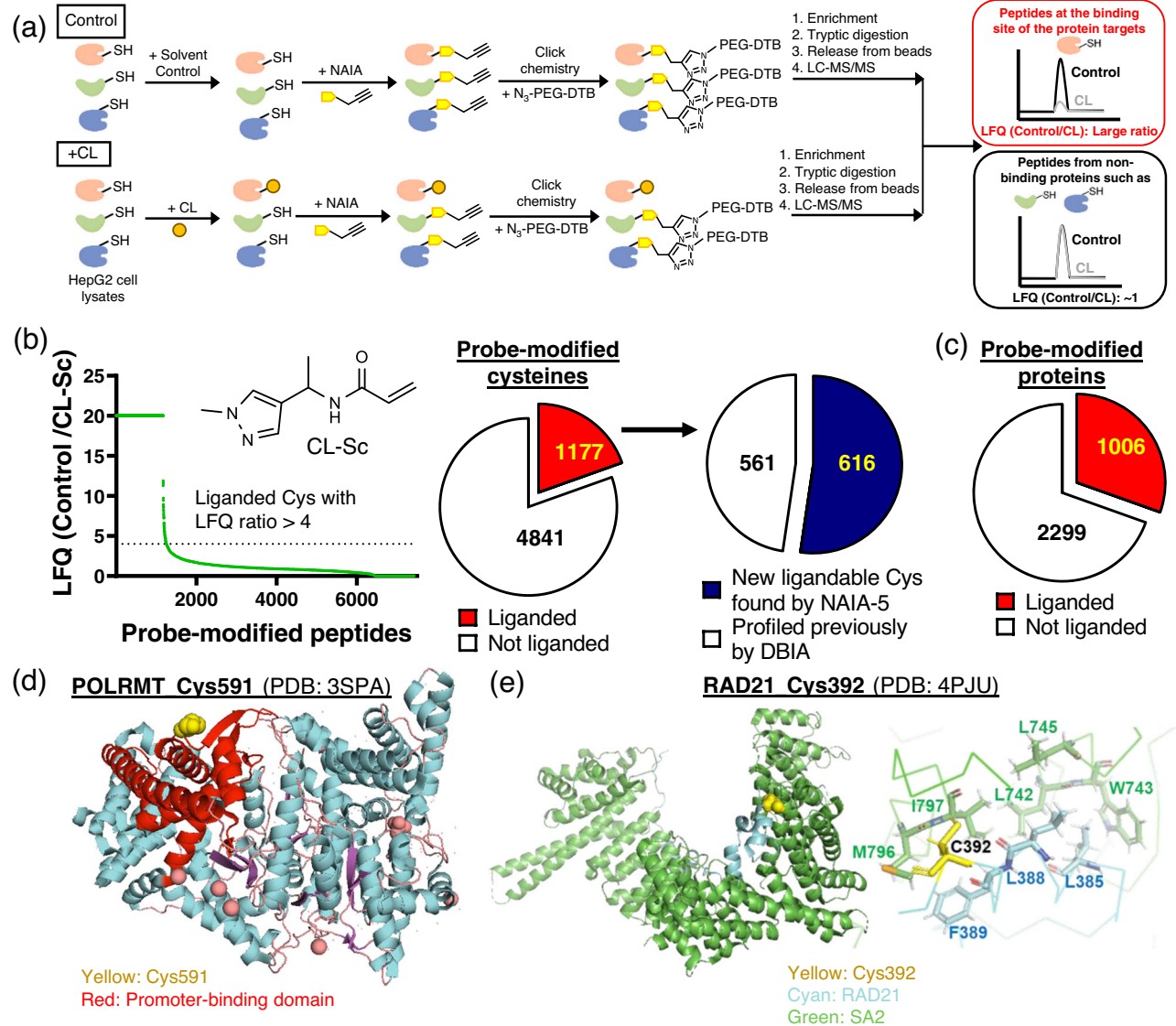

**Fig. 8 | Probe-modified cysteines by NAIA-5 are ligandable as revealed by competitive MS-based ABPP experiment using scout ligand. a** Schematic cartoon illustrating the workflow of competitive MS-based ABPP experiment to determine protein targets of covalent ligands, as read out by the large LFQ (control/treated samples). Created using ChemDraw and MS Powerpoint. **b** The scout ligand, CL-Sc, shows binding onto 1177 cysteines probed by NAIA-5, i.e., these cysteines are ligandable. Among them, 616 are uniquely profiled by NAIA-5 and have not been found in the reported study of DBIA. **c** The number of probe-modified proteins being liganded by CL-Sc. **d** A liganded Cys591 on POLRMT is located close to the promoter-binding domain; PDB ID: 3SPA. **e** Left: A ligandable Cys392 on RAD21 is located at the interface of RAD21 and SA2. Right: zoom-in image showing Cys392 and other key residues at the binding interface of RAD21 and SA2. PDB ID: 4PJU.

and proteins, respectively, were liganded by CL-Sc (Fig. 8b, c and Supplementary Data 7). Over 50% of these liganded cysteines have not been identified in the reported work of DBIA[56], and a number of them are located on cancer drivers[61], highlighting the potential of these cysteines as hotspots for therapy. Notably, POLRMT is an RNA polymerase that is essential for the transcription of mitochondrial DNA into RNA. The identified ligandable Cys591 is positioned right next to the promoter-binding domain (Fig. 8d)[62] and hence should be a good target for covalent ligands to modulate promoter binding and transcription of mitochondrial genome. Another ligandable Cys on RAD21, Cys392, is located at the interface of RAD21 and SA2, where they interact and form cohesin complexes to maintain genome integrity (Fig. 8e). D793K mutation of SA2, which is close to the Cys392 of RAD21, has been reported to abolish binding between RAD21 and SA2[63]. Covalent targeting of the ligandable Cys392 to induce steric clash could be a good strategy to disrupt RAD21-SA2 interactions.

These two representative examples highlight the potential of NAIA to identify new hotspots for therapy.

**Identification of covalent ligands targeting ligandable cysteines**
We integrated NAIA into a competitive gel-based ABPP experiment (Supplementary Fig. 21), and attempted to screen out hit compounds with good binding affinities for ligandable cysteines in HepG2 cell lysates from our small library of covalent ligands (Supplementary Fig. 22). In this experiment, a strong fluorescence signal was detected from samples treated with 0.1 μM of NAIA-5, while 1 μM of IAA is required for getting a reasonably good signal-to-noise ratio, suggesting a higher sensitivity of NAIA-5 as the activity-based probe for ligandable cysteines in the screening experiment. NAIA-5 enables the identification of a number of covalent ligands showing good bindings with proteins in HepG2 cell lysates, as indicated by the decrease in in-gel fluorescence intensity as compared to the DMSO control

(Supplementary Fig. 21). Interestingly, the hit compounds identified by NAIA-5 are quite different from IAA (Supplementary Fig. 21; hit compounds are labeled in red). More acrylamide compounds were found to be hits in the experiment using NAIA-5 while most of the hit compounds identified by IAA are covalent ligands with chloroacetamide warhead (Supplementary Fig. 21). This can be attributable to the preferential identification of hit compounds from the probe with the same reactive warhead, as supported by similar results in literature[13,19]. It is noteworthy that currently there is no promising acrylamide-containing probe developed for cysteine profiling, while many cysteine-reactive therapeutic agents with good bioactivity and efficacy are indeed acrylamide-containing compounds[22,64,65]. Successful development of a good acrylamide-containing cysteine probe, such as NAIA, is anticipated to facilitate research on cysteine-reactive compounds for translational and clinical applications.

To demonstrate the ability of NAIAs to identify new binding cysteines and proteins of covalent ligands, we applied competitive MS-based ABPP experiments to investigate one of the promiscuous covalent ligands, CL1 (Fig. 9). 79 peptides were found to show LFQ ratios (LFQ of control sample/LFQ of CL1-treated sample) larger than 5, indicating bindings of CL1 onto these cysteines and proteins. Interestingly, 43% of these cysteines were found to be new ligandable sites, and 10% of the cysteines were on newly profiled proteins (Fig. 9b and Supplementary Data 8), as compared to the reported results of DBIA[56].

GO analysis of the identified protein targets of CL1 revealed a significant association with cell cycle and cell division (Fig. 9c). Cell cycle analysis by flow cytometry on HepG2 cells treated with CL1 showed $G_1$-phase cell cycle arrest (Fig. 9d, e), in alignment with the GO functional analysis. Among the protein targets of CL1, Rac1 is a known cancer driver[61] and has been reported to modulate the cell cycle[66,67]. This prompted us to investigate the effect of CL1 on Rac1 biology through covalent targeting of the Cys178. We first confirmed in vitro binding of CL1 with purified human Rac1 protein by gel-based ABPP experiment (Supplementary Fig. 23). Then, CL1 was found to disrupt in vitro interactions between Rac1 and the guanine nucleotide exchange factor TIAM1, which is important for activation of Rac1 (Fig. 9f). In consistent with this result, HepG2 cells incubated with CL1 showed significant decreases in Rac1-GTP level (Fig. 9g) and pPAK which is the downstream effector of Rac1 (Fig. 9h). Confocal fluorescence imaging using FRET-Rac1 biosensor[68] also revealed a decrease in Rac1 activity upon CL1 treatment (Fig. 9i). In addition, CL1 treatment led to decreases in pRb, cyclin D1 and E2F1 levels in HepG2 cells (Fig. 9j), indicating cell cycle arrest at $G_1$ phase. Interestingly, genetic knockdown of Rac1 recused HepG2 cells from cell cycle arrest, as shown in flow cytometry (Fig. 9k, l) and western blotting experiments (Fig. 9m). This supports the effects of CL1 on cell cycle in HepG2 cells are primarily through covalent targeting of Rac1.

## Discussion

We have presented the design, synthesis, and applications of *N*-acryloylindole-alkynes (NAIAs) as a class of cysteine-reactive probes for proteome-wide cysteine profiling and imaging. Through delocalization of π electrons from the acrylamide warhead over the whole indole scaffold, this increases the electrophilicity of the acrylamide on NAIA and hence activates it for fast and selective thiol-Michael addition reaction with nucleophilic cysteine.

NAIA-4 and NAIA-5 have been utilized to profile functional cysteines in HepG2 and 231MFP cells. The identity of these new functional cysteines has been determined by MS-based ABPP experiments. A significant expansion of the pool of ligandable cysteines and proteins has been achieved, as compared to the current state-of-the-art cysteine profiling experiment by DBIA[56] and the recently reported cysteine chemoproteomics database[57], using NAIA-5 at a much lower working concentration (10 μM vs. 500 μM for DBIA). A number of these newly liganded cysteines are on proteins associated with gene expression

and regulations, as well as cancer drivers including POLRMT and RAD21 which are potential hotspots for therapy. We have also integrated NAIAs into competitive ABPP platforms and covalent ligand screening experiments. This allows us to identify hit compound CL1 which was found to target Cys178 on Rac1 covalently to mediate $G_1$-phase cell cycle arrest, as supported by flow cytometry, western blotting, biochemical assays, and siRNA knockdown experiments. It is noteworthy that CL1 is an early-hit compound and its specificity is yet to be optimized. The goal of the present study is to showcase the applicability of NAIAs to discover new ligandable hotspots which can be targeted by covalent ligands to modulate biological processes. We are currently applying NAIAs for a larger covalent ligand screening experiment, aiming at identifying potent lead compounds with specific targeting on cancer-associated proteins for treating cancers.

It is noteworthy that in MS-based chemoproteomics experiments, NAIAs have been found to work nicely at a low working concentration (10 μM), as well as with both simple experimental setups (using short LC gradient) and MudPIT for proteome-wide cysteine profiling. This highlights the good feasibility of using NAIAs for cysteine profiling experiments, especially when MudPIT is not available. NAIA also shows better performance than IAA in profiling functional cysteines in live cells. In addition, preliminary results show good compatibility of NAI-based probes with isobaric labeling reagents for quantitative analysis, further supporting the potential of using NAIs in different proteomics and MS experiments.

In addition to serving as probes for proteome-wide profiling of functional cysteines, NAIs have been successfully demonstrated for imaging dynamic changes in cellular thiol oxidative modifications. The fast thiol reaction kinetics and good cellular penetration allow NAI/NAIA to capture and image oxidized thiols in cells by confocal fluorescence microscopy. On the other hand, IAA, the widely used probe for studying thiol redox proteome, fails to monitor changes of oxidized thiol level in the imaging experiments, probably due to its slower thiol reactivity and hence incomplete capture of thiols in live cells. We have further applied NAI/NAIA couple to identify oxidized cysteines by MS experiment and profiled unique and important cysteines associated with redox biology which have not been identified by the reported redox proteomics platform using CPT. Interestingly, we found very different oxidation profiles of cysteines on important classes of proteins such as thioredoxins and glutathione peroxidases, as well as proteins associated with hydrogen peroxide metabolism in cells treated with $H_2O_2$. This suggests different modifications and sensitivity toward oxidation of these cysteines, which should be critical for governing cell fate in response to oxidative stress. All these results illustrate the good potential of NAI/NAIA for studying thiol reactivity and complex cysteine biology by microscopy imaging and MS experiments. This should provide new insights on thiol redox biology, particularly on the spatial information of modified thiols in cells and the roles of different cysteines in samples under oxidative stress or disease-relevant conditions.

We have compared the cysteine reactivity of NAIAs with some of the recently developed cysteine probes, such as methylsulfonylbenzothiazole (MSBT) and *N*-hydroxybenzimidoyl chloride. NAIA-5 still showed faster and more complete reactions with cysteines to form stable cysteine adducts. The higher cysteine reactivity of NAIAs does not comprise its good selectivity toward cysteines, as shown from in vitro LC–MS experiments and cysteine profiling of HepG2 cell lysates. NAIA-Cys adduct also ionizes well in ESI–MS, giving rise to a strong MS signal for detection. All these features allow NAIAs to function as promising probes for cysteine profiling and imaging. One potential drawback of using NAIAs in cysteine profiling is its relatively large modification on cysteine, as compared to conventional iodoacetamide-based cysteine probes. Although we found the excellent performance of NAIAs in gel-based and MS-based chemoproteomics experiments, this is anticipated that even better cysteine

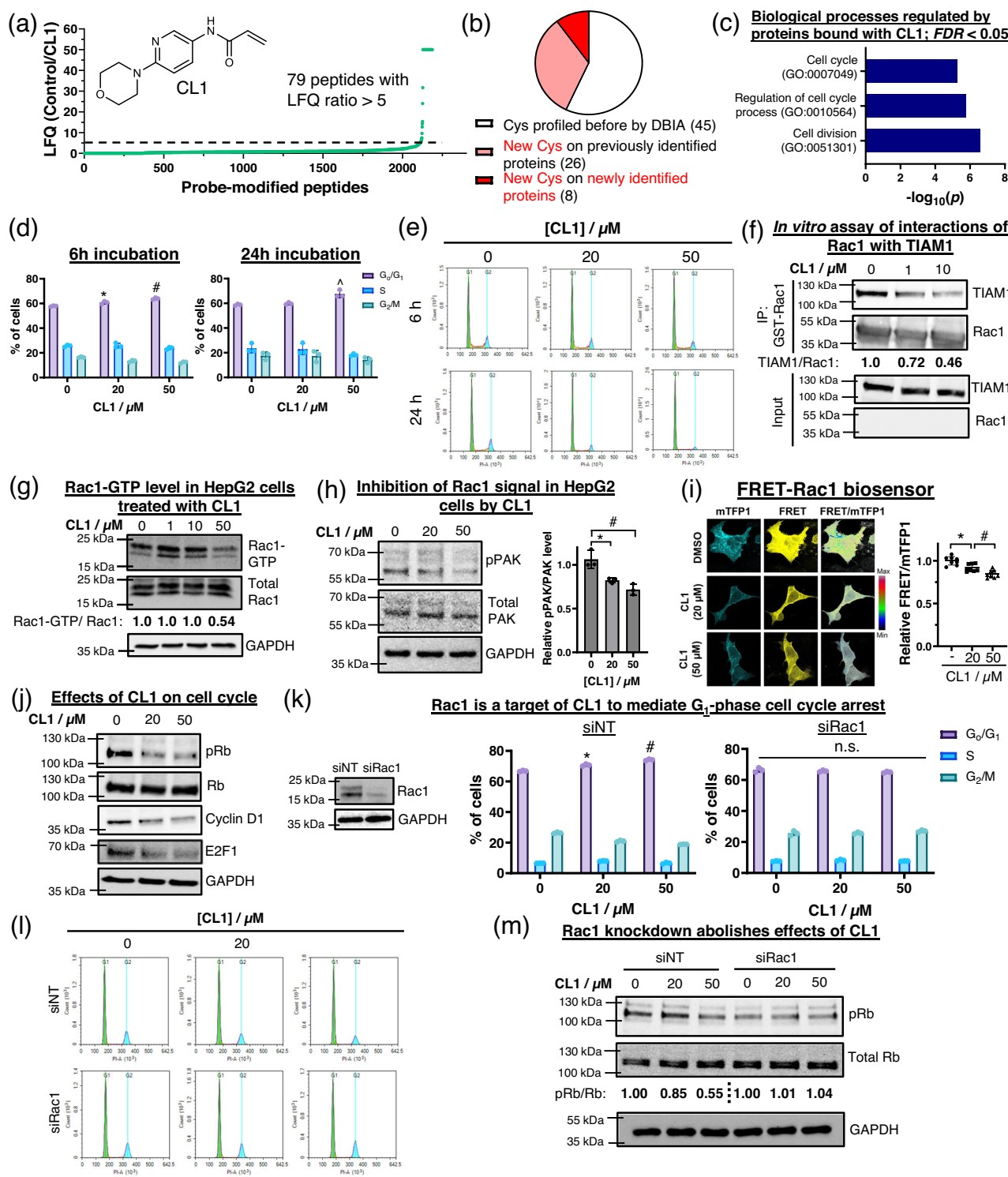

profiling can be achieved by reducing the size of the Cys modification to minimize steric hindrance on labeling cysteines. We are actively developing next-generation cysteine-reactive probes which have activated acrylamide warhead and can mediate a smaller modification on cysteines. Also, there could be concerns about the quenching of NAIAs by glutathione due to its higher thiol reactivity. Nonetheless, gel-based and MS-based ABPP experiments revealed excellent capture of functional cysteines on proteins by NAIAs over IAA, and confocal fluorescence imaging shows no significant changes in labeling functional cysteines by NAIAs in cells with higher glutathione levels. This can be partly explained by the lower preference of NAIA on labeling peptides with GCD local sequence, as found in MS experiments on live HepG2 cells, when compared to that of IAA.

In a broader chemical context, the present work represents a versatile and general strategy for developing new small-molecule compounds with faster cysteine reaction kinetics under physiologically relevant conditions through increasing electrophilicity and hence activation of acrylamide. It also opens up new avenues for the study of the chemical and biological properties of NAIs, as well as initiating the development of new functional NAIs with more attractive features for advancing their applications in imaging and chemoproteomics experiments.

**Fig. 9 | NAIA-5 enables the identification of new covalent ligand CL1 to mediate $G_1$-phase cell cycle arrest in HepG2 cells. a** Competitive MS-based ABPP experiment using NAIA-5 as the probe reveals promiscuous binding of CL1 onto 79 ligandable cysteines. **b** Analysis of the 79 liganded cysteines by CL1, revealing a number of new cysteines and proteins identified by using NAIA-5 but not DBIA as the cysteine probe. **c** Gene Ontology analysis of the biological processes involving the 79 proteins which are liganded by CL1. **d** Flow cytometry analysis of the percentage of HepG2 cells in the $G_0/G_1$, S, and $G_2/M$ phase after incubation with the indicated concentration of CL1 for 6 and 24 h respectively. **e** Representative images of the cell cycle analysis by flow cytometry. **f** Co-immunoprecipitation of GST-Rac1 shows disruption of interactions between Rac1 and TIAM1 upon incubation with CL1. **g** Rac1 activation assay indicates a decrease in Rac1-GTP level in CL1-treated HepG2 cells. **h** Western blots show decreases in pPAK levels in HepG2 cells after incubation with CL1. **i** FRET-Rac1 biosensor reveals a decrease in Rac1 activity in CL1-treated HepG2 cells. Quantified data were shown on average ± SD from $n = 8$

individual cells from 3 different biological replicates/groups. **j** CL1 treatment leads to decreases in pRb, cyclin D, and E2F1 levels in HepG2 cells. **k** Flow cytometry experiment shows no significant changes in the cell cycle of CL1-treated HepG2 cells with genetic knockdown (KD) of Rac1 by siRNA. **l** Representative images of the cell cycle analysis in (**k**) by flow cytometry. **m** pRb level in HepG2 cells with Rac1 KD is not sensitive to CL1 treatment, suggesting that Rac1 is one of the protein targets of CL1 to regulate the cell cycle. Quantified data were shown on average ± SD from $n = 3$ replicates/group. Statistical analyses were performed with unpaired two-tailed Student's $t$-tests. Statistical significance is expressed as $*P = 1.56 \times 10^{-2}$, $^{\#}P = 3.30 \times 10^{-4}$ and $^{\wedge}P = 7.88 \times 10^{-3}$ respectively as compared to the % of cells in $G_0/G_1$ in solvent control in (**d**); $*P = 1.83 \times 10^{-2}$ and $^{\#}P = 7.67 \times 10^{-3}$ respectively in (**h**); $*P = 3.14 \times 10^{-3}$ and $^{\#}P = 5.23 \times 10^{-3}$ respectively in (**i**); $*P = 1.17 \times 10^{-3}$ and $^{\#}P = 3.93 \times 10^{-5}$ respectively as compared to the % of cells in $G_0/G_1$ in solvent control in (**k**); n.s. not significant.

## Methods

### Chemical synthesis
Experimental details for chemical synthesis and characterization data of the compounds can be found in Supplementary Information.

### Cell Culture
The 231MFP cells were obtained from B. Cravatt and were generated from explanted tumor xenografts of MDA-MB-231 cells as previously described[69]. They were cultured in an L-15 medium containing 10% FBS and maintained at 37 °C with 0% $CO_2$ and were subcultured when 80% confluence was reached. HepG2 cells were cultured in DMEM medium containing 10% FBS and 1% PS and maintained at 37 °C with 5% $CO_2$ and were subcultured when 80% confluence was reached.

### Reaction kinetics with *N*-acetyl-L-cysteine methyl ester
A stock solution of the cysteine-reactive compound in DMSO was diluted by PBS/MeOH solution mixture (4:1, v/v; 500 μL), reaching a final concentration of 10 μM. A stock solution of *N*-acetyl-L-cysteine methyl ester in DMSO was then freshly prepared and added to the compound solution at a final concentration of 250 μM. The solution mixture was incubated at room temperature, and at predetermined time intervals, an aliquot of the reaction mixture (10 μL) was sent for LC-MS analysis on Waters Autopurification System using a SunFire C18 HPLC column (50 × 4.6 mm with 5 μm diameter particles, Waters). Separation was achieved by gradient elution from 5 to 100% MeCN in water (constant 0.1 vol % formic acid) over 4 min, isocratic elution with 100% MeCN (with 0.1 vol% formic acids) from 4 to 8 min, and returning to 5% MeCN in water (with 0.1 vol % formic acid) and equilibrated for 2 min. Selected ion chromatograms, with m/z corresponding to the molecular ion of the compound and/or the adduct, were extracted and the data was analyzed using MassLynx™ software by calculating the area under the curve. The peak area was then calibrated to the concentration of the compound/adduct by a set of solution mixtures containing known concentrations of the compound/adduct.

### In-gel fluorescence for visualizing Cys labeling in vitro
HepG2 or 231MFP cells were lysed by probe sonication in DPBS, and cell debris was removed by centrifugation at 1,000*g* for 5 min. The protein concentration of the lysates was determined by bicinchoninic acid (BCA) assay, and the lysates were then diluted to 2 mg/mL by DPBS. 50 μL of the lysates were incubated with indicated concentrations of NAIA-4, NAIA-5, IAA or MIA for indicated time interval at room temperature. A master mix for CuAAC was prepared from azide-fluor 545 (5 mM), copper(II) sulfate (9.5 mM), TBTA (1 mM), and freshly prepared TCEP (50 mM), and added to the lysates with the final concentrations of azide-fluor 545, copper(II) sulfate, TBTA, and TCEP in the solution mixture at 25 μM, 1 mM, 100 μM and 1 mM, respectively. The solution was incubated in the dark at room temperature with shaking for 1 h, and then the reaction was quenched with 4× reducing

Laemmli SDS sample loading buffer (Alfa Aesar) and heated at 90 °C for 5 min. Samples were then separated by molecular weight on pre-cast 4–20% Tris-Glycine Plus gels (Thermo Scientific) and scanned by ChemiDoc MP (Bio-Rad Laboratories, Inc) for measuring in-gel fluorescence. After that, the gel was washed twice with MilliQ water and then incubated with SimplyBlue™ SafeStain (ThermoFisher Scientific; LC6060) for 1 h with gentle shaking. The staining solution was discarded, and the gel was washed with MilliQ water for 1 h with gentle shaking, replaced with new MilliQ water, and washed again for 1 h. The gel was then scanned by ChemiDoc MP for imaging the blue staining.

### In-gel fluorescence for visualizing Cys labeling in live cells
231MFP or HepG2 cells were grown in 10-cm plates in a complete medium. At ca. 80% confluency, the cells were treated with DMSO solvent control, NAIA-4, NAIA-5, IAA, or MIA at indicated concentrations in a complete medium for an indicated time interval. The cells were then washed by DPBS and lysed by probe sonication in DPBS. Cell debris was removed by centrifugation at 1,000*g* for 5 min. The protein concentration of the lysates was determined by BCA assay, and the lysates were diluted to 2 mg/mL by DPBS. 50 μL of the lysates were labeled with azide-fluor 545 by CuAAC according to the procedures described above for the cell lysate experiments. The samples were then added with 4× reducing Laemmli SDS sample loading buffer, boiled at 90 °C for 5 min, and separated by molecular weight on pre-cast 4–20% Tris-Glycine Plus gels and scanned by ChemiDoc MP for measuring in-gel fluorescence. After that, the gel was stained with SimplyBlue™ SafeStain as described above, washed, and scanned by ChemiDoc MP for imaging the blue staining.

### HepG2 cells labeled by NAIA-5 for imaging experiments
HepG2 cells were plated on an 8-well Nunc Lab-Tek chambered slide system (ThermoFisher Scientific; 177402) and allowed to grow in complete medium at 37 °C with 5% $CO_2$ to ca. 70% confluency. The cells were incubated with NAIA-5 and IAA (10 μM) respectively in complete medium for the indicated time interval, washed with PBS, and then fixed by 4% paraformaldehyde in PBS at room temperature for 15 min. After washing with PBS, the fixed cells were permeabilized by PBS with 0.5 vol% Triton X-100 at room temperature for 30 min. The cells were then washed and incubated with a master mix solution containing CuSO4, THTPA, azide-fluor 545, and sodium ascorbate at final concentrations of 100, 500, 20, and 5000 μM, respectively. After incubation in the dark at room temperature for 1 h, the cells were washed with PBS and stained by Hoechst 33342 in PBS (final concentration = 8.2 μM) at room temperature for 15 min. The cells were washed thrice with PBS and then imaged by confocal fluorescence microscopy.

### Imaging free thiol level in HepG2 cells by NAIA-5
HepG2 cells were plated on an 8-well Nunc Lab-Tek chambered slide system and allowed to grow in complete medium at 37 °C with 5% $CO_2$

to ca. 70% confluency. The cells were then pretreated with solvent vehicles or $H_2O_2$ (0.5 or 1 mM), or a mixture of $H_2O_2$ (1 mM) and NAC (5 mM) in a complete medium for 15 min at 37 °C. The cells were washed with PBS, and incubated with NAIA-5 or IAA (10 μM) in a complete medium for 10 min at 37 °C. After that, the cells were fixed, labeled by azide-fluor 545, and then imaged in PBS by confocal fluorescence microscopy according to the aforementioned methods.

### Imaging free thiol in HepG2 cells with elevated GSH level by NAIA-5

HepG2 cells were plated on an 8-well Nunc Lab-Tek chambered slide system and allowed to grow in complete medium at 37 °C with 5% $CO_2$ to *ca.* 70% confluency. The cells were then pretreated with solvent vehicles or L-cysteine (300 μM), which is a precursor for cellular GSH synthesis, in a complete medium for 16 h at 37 °C. The cells were washed with PBS, and incubated with NAIA-5 or IAA (10 μM) in a complete medium for 10 min at 37 °C. After that, the cells were fixed, labeled by azide-fluor 545, and then imaged in PBS by confocal fluorescence microscopy according to the aforementioned methods.

### Imaging oxidized thiols in HepG2 cells by NAI/NAIA couple

HepG2 cells were plated on an 8-well Nunc Lab-Tek chambered slide system and allowed to grow in complete medium at 37 °C with 5% $CO_2$ to ca. 70% confluency. The cells were then pretreated with solvent vehicles or $H_2O_2$ (0.5 or 1 mM), or a mixture of $H_2O_2$ (1 mM) and NAC (5 mM) in a complete medium for 15 min at 37 °C. The cells were washed with PBS, and incubated with NAI compound **3b** (50 μM) in a complete medium for 10 min at 37 °C. After that, the cells were washed with PBS and fixed with 4% paraformaldehyde in PBS at room temperature for 15 min. The fixed cells were washed with PBS, permeabilized by PBS with 0.5 vol% Triton X-100 at room temperature for 30 min, washed again with PBS, and incubated with TCEP (1 mM) in PBS at room temperature for 1 h. The cells were then incubated with NAIA-5 (10 μM) in PBS at room temperature for 1 h, followed by washing with PBS and incubation with a master mix solution containing CuSO4, THTPA, azide-fluor 545, and sodium ascorbate at final concentrations of 100, 500, 20 and 5000 μM respectively. After incubation in the dark at room temperature for 1 h, the cells were washed with PBS and stained by Hoechst 33342 in PBS (final concentration = 8.2 μM) at room temperature for 15 min. The cells were washed thrice with PBS and then imaged in PBS by confocal fluorescence microscopy.

### Confocal fluorescence microscopy imaging and data analysis

Confocal fluorescence microscopy imaging was performed with a Zeiss laser scanning microscope 880 with a 20× water-immersion objective lens using ZEN 2.3 (Black Version) software (Carl Zeiss) with 3× magnification zoom-in. Hoechst 33342 was excited with a 405 nm diode laser, and emission was collected on a META detector between 371 and 507 nm. Fluor 545 was excited by a 561 nm diode laser and emission was collected on a META detector between 576 and 683 nm. Image analysis was performed by use of ImageJ. A region of interest (ROI) was created around the individual cells, and cellular fluorescence intensity was measured. The reported average cellular fluorescence intensity was determined by averaging the measured intensity from 30 different cells from 3 different biological replicates/groups. Statistical analyses were performed with a two-tailed Student's *t*-test (MS Excel).

### MS-based ABPP experiments on HepG2 cell lysates

HepG2 cells were lysed in PBS by sonication. After BCA assay and protein normalization, HepG2 cell lysates in PBS (2 mg/mL, 2 mL) were incubated with NAIA-5 and IAA (10 μM), respectively, at room temperature for 1 h with vortexing. Then, the samples were incubated with a master mix solution for CuAAC reaction, containing $CuSO_4$, TBTA, DTB-PEG-azide, and TECP (final concentrations are 1 mM, 100 μM, 100 μM, and 1 mM, respectively). After incubation at room temperature for 1 h

with vortexing, pre-chilled acetone (12 mL) was added to the samples, and proteins were allowed to precipitate out at −20 °C overnight. The samples were centrifuged at 5000*g* at 4 °C for 10 min, and the supernatant was discarded. The protein pellets labeled by NAIA-5 and IAA were then combined, and washed with pre-chilled methanol twice. The protein pellets were then re-dispersed in 1.2% SDS in PBS (w/v), followed by heating at 80 °C for 5 min. Any insoluble solids were discarded by centrifugation at 6500*g* for 5 min, and the supernatant was transferred to PBS solution containing Pierce™ Streptavidin Agarose beads (20349; Thermo Scientific) with a final concentration of SDS equal to 0.2% (w/v). The samples and beads were incubated at 4 °C with rotation overnight. The beads were then washed with PBS and water, and re-dispersed in 6 M urea in PBS. The samples were reduced by TCEP (1 mM) at 65 °C for 20 min, followed by alkylation with iodoacetamide (18 mM) at 37 °C for 30 min in the dark. The beads were then spun down by centrifugation at 1400*g* for 2 min, washed with PBS, and re-suspended in 2 M urea in PBS. The proteins on the beads were then digested by sequencing grade trypsin (Promega) at 37 °C overnight. After tryptic digestion, the beads were spun down by centrifugation at 1400*g* for 2 min and the supernatant was discarded. The beads were washed thrice with PBS and thrice with water, followed by the addition of elution buffer solution (MeCN/$H_2O$, 1:1, v/v; with 0.1% formic acid) to the beads and incubation for 5 min at 37 °C. The probe-modified peptides were eluted out from the beads and the supernatant was collected. The beads were incubated with a new elution buffer solution, spun down and the supernatant solution was combined with the previous elution buffer solution. The combined solution containing the eluted probe-modified peptides was dried and desalted by C18 Stage tips, and the peptides (200 ng) were sent for LC–MS/MS analysis using Aurora C18 UHPLC column (75 μm i.d. × 25 cm length × 1.6 μm particle size; IonOpticks, Australia) coupled to timsTOF Pro mass spectrometer (Bruker).

Chromatographic separation was carried out using buffer A (98:2 water:acetonitrile, 0.1% formic acid) and B (acetonitrile, 0.1% formic acid) with the gradient from 98% buffer A to 30% buffer B at a flow rate of 300 nL/min over 100 min, followed by an increase from 30 to 44% buffer B over 5 min, an increase to 95% buffer B in 0.5 min, an isocratic gradient of 95% buffer B over 8.5 min, a decrease of buffer B to 2% in 0.5 min and then an isocratic gradient of 2% buffer B for 5.5 min. MS data was collected over a m/z range of 100–1700, and an MS/MS range of 100–1700. During MS/MS data collection, each TIMS cycle was 1.1 s and included 1 MS + an average of 10 PASEF MS/MS scans.

The data were searched against the UniProt human database using MaxQuant v2.0.3.0, specified with trypsin digestion (allowed up to 3 missed cleavages) and cysteine carbamidomethylation (+57.02146) as a static modification. The search also allowed up to 5 variable modifications for methionine oxidation (+15.99491), *N*-terminal acetylation (+42.01056), cysteine modification by NAIA-5 (+681.38500) or cysteine modification by IAA (+494.32167). The peptide false discovery rate (FDR) was set to 1%.

The data was also searched by unbiased open search using MSFragger v3.7, specified with precursor mass tolerance ranging from −150 Da to +1000 Da. The amino acid selectivity of NAIA-5 and IAA were determined by the unannotated mass-shifts of +681.38500 ± 100 and +494.32167 ± 100, respectively, in the global profile analysis.

### MS-based ABPP experiment on live cells incubated with NAIA

HepG2 cells were treated with NAIA-5 and IAA (100 μM), respectively, in a serum-free medium for 30 min at 37 °C. After that, the cells were washed with PBS and lysed in PBS by sonication. The samples were then prepared, run on a timsTOF Pro mass spectrometer, and analyzed using the aforementioned protocol for the cell lysate experiment.

### Competitive MS-based ABPP experiments on CL-Sc and CL1

HepG2 cell lysates in PBS (2 mg/mL, 2 mL) were first treated with DMSO control or CL-Sc (500 μM)/CL1 (50 μM) at room temperature for 1 h.

The samples were then incubated with NAIA-5 (10 μM) at room temperature for 1 h with vortexing before CuAAC, followed by acetone precipitation. Without pairing of different groups of samples, the samples were washed with methanol, and subjected to the same preparation protocol and analysis on the timsTOF Pro mass spectrometer as described for the MS-based ABPP experiments on HepG2 lysates.

The data were searched against the UniProt human database using MaxQuant v2.0.3.0, specified with trypsin digestion (allowed up to 3 missed cleavages) and cysteine carbamidomethylation (+57.02146) as a static modification. The search also allowed up to 5 variable modifications for methionine oxidation (+15.99491), N-terminal acetylation (+42.01056), or cysteine modification by NAIA-5 (+681.38500). The peptide FDR was set to 1%. In the data analysis, peptides with non-zero LFQ intensity for all the 2 replicate runs in the control group and zero LFQ intensity for all the 2 runs in the treated group were assigned LFQ ratios of 20 (for the experiment of CL-Sc) or 50 (for the experiment of CL1). The LFQ ratios for other peptides were calculated by (average LFQ intensity of control samples)/(average LFQ intensity of treated samples), and those peptides with invalid values were filtered out.

### NAI/NAIA couple to study cysteine oxidations by MS

HepG2 cells were treated with solvent vehicle, $H_2O_2$ (1 mM) or $H_2O_2$ (1 mM)+NAC (5 mM) in a serum-free medium for 15 min at 37 °C. The cells were then incubated with NAI compound **3b** (50 μM; 10 min) to label-free thiols in live cells, followed by cell lysis and TCEP (1 mM) reduction for 1 h. After acetone precipitation at −20 °C overnight, the proteins were labeled by NAIA-5 (100 μM) in PBS. The samples were then prepared and run on a timsTOF Pro mass spectrometer following the same procedures in the competitive MS-based ABPP experiments.

The data were searched against the UniProt human database using LFQ-MBR workflow in MSFragger 3.7, specified with trypsin digestion (allowed up to 2 missed cleavages) and cysteine carbamidomethylation (+57.02146) as a static modification. The search also allowed up to 5 variable modifications for methionine oxidation (+15.9949), N-terminal acetylation (+42.0106), or cysteine modification by NAIA-5 (+681.3850). For the peptides with zero LFQ intensity for all the 2 replicate runs in the control group and non-zero LFQ intensity for all the 2 runs in the treated group, they were assigned for an LFQ ratio of 50. On the other hand, peptides with non-zero LFQ intensity for all the 2 replicate runs in the control group and zero LFQ intensity for all the 2 runs in the treated group were assigned LFQ ratios of 0.01. The LFQ ratios for other peptides were calculated, and peptides with invalid values were filtered out. UniProt feature Key annotations and Gene Ontology Biological Process (GOBP) of the identified cysteines were determined using Perseus v2.0.9.0.

### Western blot analysis

Protein samples were separated by SDS-PAGE in 10% TGX Stain-Free polyacrylamide gels and transferred onto nitrocellulose membrane. The membrane was blocked with 5% bovine serum albumin (BSA) in Tris-buffered saline with 0.1% Tween 20 (TBST) for 1 h at room temperature. The membranes were then incubated with primary antibodies in 5% BSA/milk in TBST with 1% $NaN_3$ (1:1000 dilution, except anti-pPAK antibody with 1:500 dilution) at 4 °C overnight, followed by horseradish-peroxidase coupled (HRP) secondary antibodies (CST #7074 and 7076; 1:2000 dilution) at room temperature for 1 h. The membranes were developed by the LumiGlo reagent (CST #7003) and imaged by ChemiDoc MP.

### Recombinant protein preparation

GST-Rac1 construct (Addgene #12977) was expressed in BL21 *Escherichia coli* Bacteria were pelleted by centrifugation at 5000*g* for 10 min, and lysed by pipetting up and down till homogenization in B-PER Reagent (Thermo #78243) with 15 min incubation at room temperature. After centrifugation at 16,000*g* for 20 min, the supernatant containing the recombinant protein was collected and purified by incubation with glutathione-agarose beads (Thermo #16100) for 1 h at 4 °C. The protein was then eluted from the beads using elution buffer (50 mM Tris, 150 mM NaCl, 10 mM glutathione, pH 8.0), and subjected to dialysis to remove excess glutathione. The purity of the protein has been examined by SDS-PAGE and Coomassie Blue staining.

mNeonGreen-tagged human Tiam1 (DH domain) construct (Addgene #172102) was expressed in HEK293T cells by transient transfection. HEK293T cells in a 10-cm dish were grown to 30% confluency and transfected with the construct (25 μg) with Lipofectamine 2000 (ThermoFisher Scientific) for 48 h according to manufacturer's instruction. Afterward, cells were harvested by scraping and lysed by probe sonication in PBS to obtain lysates containing mNeonGreen-TIAM1 recombinant protein.

### Flow cytometry analysis on the cell cycle of HepG2 cells

HepG2 cells were plated on 6-well plates (800,000 cells) and cultured overnight in a complete medium at 37 °C with 5% $CO_2$. After treatment with CL1 or solvent control at indicated concentration and duration, cells were harvested by trypsin digestion, washed in PBS, and fixed in 70% cold ethanol on ice for 30 min. Cells were centrifuged for 5 min at 250 *g* and resuspended in staining buffer (PBS with 0.05% Triton X-100) with propidium iodide (50 μg/mL) and RNase (100 μg/mL) for 30 min at 37 °C in dark. The cells were then filtered by 40 μm strainer and analyzed by flow cytometry, with a representative gating strategy shown in Supplementary Fig. 24. NovoCyte Advanteon BVYG analyzer was used to acquire cell events and data were processed with NovoExpress software.

### Rac1 activation assay

Rac1 activation was measured by Rac1 Activation Assay Biochem Kit (Cytoskeleton # BK035). HepG2 cells were treated with CL1 at the indicated concentration for 24 h and then harvested by lysis buffer (10 mM β-glycerol phosphate, 10 mM $Na_4P_2O_7$, 40 mM HEPES, 4 mM EDTA, supplemented with Protease and Phosphatase inhibitor (ThermoFisher Scientific #A32961)). The protein samples were subjected to pull down with the Cdc42/Rac1 interactive binding (CRIB) domain of PAK1 kinase according to the manufacturer's instruction. Rac1-GTP and total Rac1 were detected by Western blot.

### Rac1-GEF interaction assay

GST-tagged Rac1 (10 μg) in binding buffer (200 μL; 50 mM HEPES, 50 mM NaCl, 5 mM MgCl2, pH 7.4) were incubated with 15 μL of suspended glutathione-agarose beads (Thermo #16100) with head-to-head rotation at 4 °C for 1 h. After immobilization, beads were washed twice. CL1 was then added at indicated concentration in binding buffer and incubated for an additional 1 h at room temperature. Cell lysates containing mNeonGreen-tagged TIAM1 (200 μL; 4 mg/mL) were added to the washed beads. After 4 h of incubation, beads were washed, and proteins were eluted by boiling at 95 °C in an SDS sampling buffer. Samples were subjected to SDS-PAGE and western blot analysis.

### FRET Rac1 biosensor to image cellular Rac1 activity

HepG2 cells were plated on an 8-well Nunc Lab-Tek chambered slide system and allowed to grow in complete medium at 37 °C with 5% $CO_2$ to *ca.* 50% confluency. The cells were then transfected with the FRET Rac1 biosensor construct (Addgene #66110) and Lipofectamine 2000 (ThermoFisher Scientific) for 48 h according to the manufacturer's instruction. The transfected cells were incubated with CL1 at indicated concentrations for 24 h, washed with PBS, and imaged in DPBS by confocal fluorescence microscopy. The cells were excited at a 458 nm diode laser, and mTFP1 emission was collected on a META detector between 470 and 495 nm, while the FRET channel was collected on a META detector between 516 and 550 nm. Image analysis was performed by use of ImageJ, with FRET/mTFP1 images generated by the

Ratio Plus plugin of ImageJ. For quantification, an ROI was created around the individual cell, and cellular FRET/mTFP1 was measured. The reported FRET/mTFP1 was determined by averaging the measured intensity from 8 different cells from 3 different biological replicates/groups. Statistical analyses were performed with a two-tailed Student's *t*-test (MS Excel).

### siRac1 experiments
HepG2 cells in a 6-well plate (300,000 cells) were transfected with siRac1 (Dharmacon #L-003560-00) or siNT (Dharmacon # D-001810-10) using DharmaFect Reagent 1 (Dharmacon #T-2005-01) according to manufacturer's instruction. After transfection for 48 h, the cells were treated with CL1 for 24 h. Afterward, cells were harvested and assayed by western blot or flow cytometry as described above.

### Statistics and reproducibility
For Figs. 5b and 9j, they were representative images from 3 replicate experiments/groups and similar results were found in all these experiments. For Fig. 9f, m, they were representative images from 2 replicate experiments/groups and similar results were found in all these experiments. Figure 9g and Supplementary Fig. 21a, were results from a single experiment. Note that the gel-based ABPP experiment in Supplementary Fig. 21a was a screening experiment, and the hit compound CL1 has been further validated by a dose-dependent experiment with two replicates/group, as shown in Supplementary Fig. 23.

## Data availability
The mass spectrometry proteomics data have been deposited to the ProteomeXchange Consortium via the PRIDE[70] partner repository with the dataset identifier PXD041264. The analyzed MS data are provided in Supplementary Data 1–8. The structural data of POLRMT and RAD21 were downloaded from PDB 3SPA and 4PJU respectively. The human proteome for proteomics analysis was downloaded from UniProt (Proteome identifier: UP000005640). We have also provided raw data and images, as well as data points for all of our plots with this paper as Source data. Other data are available by written request to the authors.

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

## Acknowledgements

C.Y.-S.C. acknowledges support from the Early Career Scheme (27315922) and Theme-based Research Scheme (T12-716/22-R) from University Grants Committee, and Seed Fund for Basic Research for New Staff (202009185025) and Enhanced New Staff Start-up Research Grant from the University Research Committee and Li Ka Shing Faculty of Medicine, The University of Hong Kong. This work was also supported by the Center for Oncology and Immunology under the Health@InnoHK Initiative funded by the Innovation and Technology Commission, The Government of Hong Kong SAR, China. T.-Y.K. and H.L. acknowledge the receipt of a University Postgraduate Fellowship and Postgraduate Studentship administered by The University of Hong Kong respectively. We thank for the support from the Imaging and Flow Cytometry Core and Proteomics and Metabolomics

Core, from the Center for PanorOmic Sciences (CPOS), Li Ka Shing Faculty of Medicine, The University of Hong Kong, on the confocal fluorescence microscopy imaging, flow cytometry, and MS experiments. We also thank Ms. Bonnie Yan and Mr. Ivan Lai, at the Department of Chemistry and Department of Microbiology, The University of Hong Kong, for their help in NMR experiments.

## Author contributions

D.K.N. and C.Y.-S.C. conceived the research. T.-Y.K., H.L., and C.Y.-S.C. designed and performed the experiments. T.-Y.K., H.L., and C.Y.-S.C. analyzed the data. T.-Y.K., H.L., and C.Y.-S.C. wrote the paper.

## Competing interests

A patent application from C.Y.-S.C., H.L., and T.-Y.K. has been filed for NAIA, and the US patent application number is 18/307,905. D.K.N. is a co-founder, shareholder, and scientific advisory board member for Frontier Medicines and Vicinitas Therapeutics. D.K.N. is a member of the board of directors for Vicinitas Therapeutics. D.K.N. is on the scientific advisory board of The Mark Foundation for Cancer Research, MD Anderson Cancer Center, Photys Therapeutics, Apertor Pharmaceuticals, Oerth Bio, and Chordia Therapeutics. D.K.N. is also an Investment Advisory Board Member for Droia Ventures and a16z.
