## [Peer Review File · Nature Communications]

N-Acryloylindole-alkyne (NAIA) enables imaging and profiling new ligandable cysteines and oxidized thiols by chemoproteomicsREVIEWER COMMENTS

Reviewer #1 (Remarks to the Author):

This manuscript by Lai et al. develops and applies a clickable variant of acrylamide for in vitro and in vivo cysteine studies. While this NAIA probe has some modest advantages over existing chemical probes, the overall impact is modest and the scientific advances are incremental. In addition, key experiments and comparisons are not performed.

1. The primary advantage of NAIA that the authors focus on is reactivity. However, the authors compare NAIA to iodoacetamide, despite it is being well known that NEM is a much faster and more specific reagent than iodoacetamide (Analytical Biochemistry 358 (2006) 171–184, <https://doi.org/10.1016/j.bbapap.2012.08.002> and many more). There is also a clickable, alkyne version of NEM commercially available. A comparison to NEM is needed to evaluate the improvement in reaction kinetics of NAIA to contemporary cysteine labeling reagents.
2. The authors claim that NAIA can uncover a 'unique pool of cysteines that can be liganded as new drug opportunities. However, they make this claim based on comparison to labeling with IAA at a concentration they admit is suboptimal. To claim that these sites are uniquely labeled by NAIA requires comparison to optimal levels of IAA or other studies that have used IAA-based warheads for redox proteomics such as CPT6 (DOI: 10.1016/j.cell.2020.02.012) and other studies employing IAA-alkyne (<https://doi.org/10.1016/j.cell.2017.08.051>, <https://doi.org/10.1021/acscchembio.9b00424>).
3. The authors note that a problem of IAA is off target reactivity with cysteine sulfenic acids, but do not evaluate the reactivity of NAIA with this group. Also, many off target effects of IAA were identified using unbiased proteomics and detecting other residues were modified. The authors did not consider modifications of other residues by NAIA in their proteomic data analysis and thus might have missed off target labeling.
4. While the cell cycle arrest by CL1 is interesting, binding competition experiments are now commonplace and the data provide no mechanistic understanding of how CL1 works. What is the functional target, or what is the mechanistic link to cell cycle proteins?
5. The authors claim that "Trapping free thiols in live cells should be preferable" to in vitro labeling. But, it is well known that cysteine alkylation reagents are highly toxic, with general agreement that blocking critical thiols in enzymes with disrupt cell function, blocking cysteines important for antioxidant capacity will induce oxidative stress which – in turn – will lead to cysteine oxidation that will prevent NAIA from binding. In short, the debate between live cell labeling and in vitro labeling is nuanced, complex, and discussed in many reviews and opinion pieces and it is not self-evident that trapping free thiols in live cells is preferable. It is notably that while NAIA is more reactive than IAA it is shown to be less toxic in Fig 5C. Can the authors reconcile this paradox?

Reviewer #2 (Remarks to the Author):

In their manuscript, Lei et al. Have developed N-acryloylindole-alkynes (NAIAs) as promising new probes for the proteome-wide profiling of cysteine residues, a technology that has vast implications for understanding fundamental biology and in drug discovery. Let me start by saying, that I am normally skeptical, if we need more probes for cysteine monitoring in these technologies as many of these probes already exists. Nevertheless, this work stands out in various aspects: 1) The authors show advantages of the new probe in various scenarios, not only over traditionally used iodoacetamide alkyne (IAA), but also over some other of the newest generation of probes. 2) The authors convincingly show applicability of the probes in living cells for microscopy and proteomics with low

toxicity, which is very hard to achieve so far. 3) Maybe most importantly, the probe (to my knowledge for the first time in such a broad profiling probe) uses a terminal acrylamide as the reactive group. This has important implications, as acrylamides are the reactive group used mostly in clinically approved drugs and, therefore, being able to profile this compound class with a probe with the same reactive group could have huge implications for drug discovery. I, here, agree with the authors that state: "It is noteworthy that currently there is no promising acrylamide-containing probe developed for cysteine profiling, while many cysteine-reactive therapeutic agents with good bioactivity and efficacy are indeed acrylamide-containing compounds. Successful development of a good acrylamide-containing cysteine probe, such as NAIA, is anticipated to facilitate research on cysteine-reactive compounds for translational and clinical applications.". I am, therefore, convinced that this probe will find a lot of important applications in chemoproteomics and is interesting to the broad readership of Nature Communications.

The research is performed to very high standards and covers a lot of different technologies. The authors synthesize the probes, characterize them with isolated amino acids as well as in vitro and in cells both in gel-based experiments, in MS-based proteomics and in microscopy studies. They perform clear and fair comparisons to state-of-the-art technologies to show where their probes are superior. The authors make all proteomics data available through the PRIDE repository, which will enable others to quickly explore the power of the technology.

I am convinced, that this is an excellent study that should be published in Nature Communications. I only have a few minor comments that should be considered.

1) The authors throughout the paper define any cysteine they detect as "ligandable". Nevertheless, the field and most prominently Ben Cravatt's group has defined a ligandable cysteine to be engaged by a competitor at high stoichiometry (usually at least 75%). This makes a difference as the here detected cysteines can be engaged only to a low degree by NAIA's, which is fine for detection, but does not allow to infer that they can also easily be made amenable to covalent drug discovery. In that sense, the authors in my eyes only identify 79 ligandable cysteines here (the ones strongly competed by CL1). This should not take away from the study, but I am convinced that this wording must be harmonized with what is standard in the field.

2) The authors should provide detail, how the amino acid selectivity by proteomics has been determined.

3) On page 3: "Yet, low cysteine reactivity has been found in aqueous buffer solution for acrylamide compounds, similar to iodoacetamides." The reactivity of normal acrylamides should be much lower than iodoacetamide's, right? Maybe "in comparison to" instead of "similar to"? I see the point that the reactivity of NAIA is still higher than iodoacetamide, but iodoacetamide and "normal" acrylamide are surely not similar in reactivity.

4) On page 10, the authors use competitive gel-based ABPP to prioritize compounds for MS-based proteomics. They write: "Interestingly, NAIA-5 enables identification of a number of covalent ligands showing good bindings with proteins in HepG2 cell lysates, as indicated by the decrease in in-gel fluorescence intensity as compared to the DMSO control (Figure 8a). On the other hand, IAA also identified some hit compounds, but fewer than those by NAIA-5." It is currently very hard to see, in the gel, what they mean here. I think bands that are considered as liganding events should be marked. Also, as they make quantitative claims from these gels, how is a hit compound clearly defined here? I surely see the value of these experiments, but as only few cysteines can truly be differentiated on a gel, I would suggest to be careful with such quantitative claims.

Reviewer #3 (Remarks to the Author):

Lai and colleagues introduce an optimized class of acrylamide-based reactive cysteine probes for in situ cell labeling. These probes are demonstrated to outcompete IAA by multiple experiments in human cell cultures at low working concentrations (10-50 μ M) and its potential for covalent ligand discovery was also presented. Overall, the results are convincing, although some sections could be shortened and presented in a more intuitive way. I outline additional comments and concerns below.

Nevertheless, I find these optimized trapping versions of acrylamide of interest to the general redox community.

Major:

- Proteomics:

o In the proteomic searches you specified cysteine carbamidomethylation as a fixed modification and then on top of that the variable modifications of NAIA-5 and IAM, I assume you then accounted for the fact that the total monoisotopic mass of the adduct should be reduced by 57 Da due to the fixed IAM setting?

o How did you perform the off-target labeling? You then thus conducted a blind search? Thus performing a mass tolerant search, e.g. in your case [-150 Da;+1000Da] for instance? Personally, when introducing new chemoproteomic probes, such searches are very informative and offer you an unbiased view of your introduced adducts and their location. Currently, I am unsure if this was performed here.

o Did the labeled peptides contain any diagnostic fragment ions? Thus certain peaks resulting due to fragmentation of the relatively large NAIA group? If so, this could aid in confident identification of labeled sites.

o Why was the chemoproteomics not attempted after labeling in live cells? This would give an even better result compared to IAM labeling and showcase the benefits of your probe?

o I don't know whether isotopic labeling strategies with such probes could be discussed as a future option as well.

- Page 5-6, Can we conclude already here from these fluorescence intensity that there are proteins uniquely labeled? The intensities are of course greater, and likely there will always be unique labeled proteins to either IAM or NAIA-4/5, however chemoproteomic approaches will only deliver you real quantitative and conclusive data to support these statements. Of course the greater band intensities seem to suggest that they might target additional cysteine targets.

- Page 7, Figure 6b. I am confused by the IAM vs NAIA-5 confocal imaging. A similar comparison was done in the previous paragraph in Figure 5 and there 10 μ M IAA for 15 min (Figure 5a) did give fluorescence, and now 50 μ M IAA for 10 min (Figure 6b) did not give any fluorescence? This is due to different imaging parameters, or I overlooked something?

- As also touched upon in the discussion, I was wondering regarding the reactivity towards non-proteinaceous thiols. Could glutathione not be tested in vitro next to single amino acid residues like in terms of probe reactivity as in Figure 3e? You could then measure the probe consumption. In addition, confocal imaging experiments can be envisioned after perturbing the glutathione levels.

Minor:

- The manuscript would benefit from English proofreading.

- References often tend to be included as '.3,10-12', should be '3,10-12.'?

- Page 3, liganded by activity-based probes

- Page 3, I guess it is logical activity-based probe are a critical component in activity-based protein profiling.. Sentence could be better rephrased to give the core message that development of activity-boost profiling could aid in targeting novel cysteine ligands and new druggable hotspots.

- Page 3, 'by forming the acrylamide group using scaffold with less electron-rich nitrogen', has to be rephrased
- Page 4, The first paragraph reads to me a bit too much like a Methods section, I think some details regarding the synthesis could be omitted and a reference could be made to the Methods section (or in this case Supporting information). Abbreviations such as 'NAIa' are not introduced in the main text (was spelled out in the figure).
- Figure 2a, the reaction scheme is a bit confusing due to the 'DDQ, toluene' shift from right to left of the figure (i.e. from compounds 3a/3b to 4a/4b). Typically, you would go down and then left to keep a consecutive order.
- Page 5, why refer to Figure 1c (introduction figure) regarding the reaction of NAIa-5 with N-acetylcysteine methyl ester?
- Figure 3:
 - o Why were not the same time intervals could be used for the SIC for substrate/adduct and LC-MS? Why was the LC-MS done at 5.6 min (= 336 seconds) and 6.2 min (= 372 seconds)? For instance, in Figures S1 and S2 the SIC time intervals were identical. Of course, the reaction and thus adduct formation of NAIa-5 is convincing.
 - o It could be useful for visual interpretation to have a horizontal line at 10 μM , which should be the starting concentration/no reaction. Would also apply for Figure S1d.
 - o Why term it as a NAIa-5+Na⁺ adduct compared to NAIa-4+H⁺ adduct in Figure S1? The LC-MS peak intensities look identical, with the Na⁺ adduct being lower in intensity compared to the H⁺ adduct?
- Would the timing of the complete reaction be more accurately described by overlaying Figure S1c (NAIa-4) and Figure 3c (lower panel, NAIa-5)? Now it says the reaction of NAIa-5 was complete within 300 s, but judging from Figure 3c it happens around ~ 120 s?
- Page 5, 'NAIa-5 and IAA were capable of labeling reactive cysteines on proteins in HepG2 cell lysates.' How did you confirm it? I was initially confused as I thought Figures 4a and 4b were already performed in live cells.
- Figure S1:
 - o There is an error in the legend: "The mass spectra (MS) at 5.19 and 4.82 min confirm the identity of NAIa-5 and adduct", this should be "NAIa-4 and adduct" I believe.
 - o Error bars missing for Figure S1d, or only one replicate (not indicated in legend)?
- Figure S2: '[MBST] / μM ' for y-scale in panel c? Why ranging above the starting concentration of 10 μM ?
- Figure S3: the concentration of N-hydroxybenzimidoyl chloride can not be determined for panel c as before for Figure S1, S2 and Figure 3? Makes it difficult for a comparison of this graph to the other probes.
- Page 6, Although agreeing, I would prefer a more quantitative value than a 'huge difference' in in-gel fluorescence intensities between NAIa-5 and IAA labeling.
- Figure 4:
 - o I think Figure 4a is dispensable as Figure 4b is giving a more detailed overview of the labeling in cell lysates and indicating time ranges (which are lacking for Figure 4a). It would also shorten the text and give the main results in a more concise manner.

o Which statistical test was used to determine these p-values? Should be indicated in the legend.

- Page 6, I would not fast-forward in the text referring to Figure 5 in the first section. It makes more sense to give a conclusion at the end of the section that the differences can be attributable to both the uptake and enhanced reaction kinetics.

- Page 7, please introduce/spell out 'NAC'.

- Figure 6:

o The legend for panel a reads set-up to monitor oxidative modifications, however that only applies to panels e and f in my opinion. Regarding panels a to d you look at reduced cellular labeling due to thiol oxidation, but you do not directly use NAI-A-5 to monitor oxidized thiols (well you could say in an indirect manner).

o How was NAC applied to the cell cultures? Pre-incubate together with H₂O₂ for 15 min or? The legend is not very descriptive for panel c and d.

Revision summary of figures and supplementary data

Original manuscript	Revision	New data/changes	Reviewers' comments
Fig. 2	Fig. 1b and 1c	Fig. 1b : revise the schematic scheme	Reviewer 3, Minor #7
Fig. 3	Fig. 2	Fig. 2b : replace LC traces and the MS	Reviewer 3, Minor #9 and #11
		Fig. 2c : New MIA data and add indicating 10 μ M	Reviewer 1, Comment #1; Reviewer 3, Minor #10 and 12
		Fig. 2d : New MIA data	Reviewer 1, Comment #1
Fig. 4	Fig. 3	Fig. 3a : Label the time-point	Reviewer 3, Minor #19
		Fig. 3c and 3f : New MIA data	Reviewer 1, Comment #1
Fig. 5	Fig. 4	Fig. 4b : Change to violin plot	For formatting
Fig. 6	Fig. 5c	Fig. 5c, 5d and 5f : The quantification data is now presented using violin plot	For formatting
	Fig. 6	Fig. 6 : A new experiment using NAI/NAIA couple to identify oxidized Cys by MS experiment	Reviewer 1, Comment #2 and 5
Fig. 7	Fig. 7	Fig. 7b : Update the analysis of off-target binding with other amino acid and sulfenic acid	Reviewer 1, Comment #3; Reviewer 2, Comment #2; Reviewer 3, Comment #2
		Fig. 7h-7j : New MS data on live cell labeling experiment by NAIA-5	Reviewer 1, Comment #1, 2 and 5; Reviewer 3, Comment #4
		Fig. 7i : New data to compare our results with the current state-of-the-art cysteine profiling experiment (we also have data in Fig. 7d in the original manuscript)	Reviewer 1, Comment #2
	Fig. 8	Fig. 8a-8e : New data to show that a number of probe-modified Cys of NAIA-5 are ligandable	Reviewer 1, Comment #1; Reviewer 2, Comment #1
		Fig. 8b : New data to compare our result with the current state-of-the-art cysteine profiling experiment (We also have data in Fig. 7d in the original manuscript)	Reviewer 1, Comment #2
Fig.8c-h	Fig. 9a-e		

	Fig. 9	Fig. 9f-9m : New data to study mechanism of action of CL1 to mediate G ₁ -phase cell cycle arrest	Reviewer 1, Comment #4
Fig. S1	Fig. S1	Fig. S1c : add the line indicating 10 μ M	Reviewer 3, Minor #10
	Fig. S4	Fig. S4 : New MIA data	Reviewer 1, Comment #1
	Fig. S5	Fig. S5 : Reactivity of NAIA with GSH	Reviewer 1, Comment #1 and 5; Reviewer 3, Comment #8
Fig. S4	Fig. S6		
Fig. S5	Fig. S7		
Fig. S6	Fig. S8		
Fig. S7	Fig. S9		
Fig. S8	Fig. S10		
Fig. S9	Fig. S11		
	Fig. S12	Fig. S12 : Confocal images showing no significant change in NAIA-5 labeling in cells with higher glutathione level	Reviewer 3, Comment #8
Fig. S10	Fig. S13		
	Fig. S14	Fig. S14 : Comparison of the Cys identified by NAI/NAIA-5 couple in HepG2 cells with the reported CPT results	Reviewer 1, Comment #2
Fig. S11	Fig. S15		
Fig. S12	Fig. S16		Reviewer 1, Comment #3
Fig. S13	Fig. S17	Fig. S17 : Add the analysis of off-target binding onto other residues and sulfenic acid	Reviewer 1, Comment #3; Reviewer 2, Comment #2; Reviewer 3, Comment #2
	Fig. S18	Fig. S18 : Comparison of the Cys identified by NAIA-5 with the reported results from IAA at its optimal working conditions	Reviewer 1, Comment #2
	Fig. S19	Fig. S19 : New MS data on live cell labeling experiment by NAIA-5	Reviewer 1, Comment #1 and 5; Reviewer 3, Comment #4 and 8
Fig. S14	Fig. S20		
Fig. 8a	Fig. S21	Fig. S21 : Showing the quantification data of the gel-based ABPP experiment to indicate how lead compounds are identified	Reviewer 2, Comment #4
	Fig. S23	Fig. S23 : Gel-based ABPP experiment to investigate in vitro binding of CL1 with Rac1	Reviewer 1, Comment #4
	Fig. S24	Fig. S24 : Graphic illustration of gating strategy for FACS	

	Supplementary Data 1	MS data on profiling oxidized cysteines by NAI/NAIA-5 couple	Reviewer 1, Comment #2
Supplementary Data 1	Supplementary Data 2		
Supplementary Data 2	Supplementary Data 3		
Supplementary Data 3	Supplementary Data 4		
	Supplementary Data 5	Supplementary Data 5 : Analysis of new MS data from live cell labeling experiment by NAIA-5	Reviewer 3, Comment #4
Supplementary Data 4	Supplementary Data 6	Supplementary Data 6 : Analysis of new MS data of ligandable cysteines	Reviewer 2, Comment #1
	Supplementary Data 7		Reviewer 2, Comment #1
Supplementary data 5	Supplementary Data 8		Reviewer 2, Comment #1

Response to Reviewers' comments:

We thank the reviewers for their comments and suggestions, which can further strengthen our paper. All changes in the revised manuscript are highlighted in yellow. Our point-by-point responses are in blue font.

REVIEWER COMMENTS

Reviewer #1:

This manuscript by Lai et al. develops and applies a clickable variant of acrylamide for in vitro and in vivo cysteine studies. While this NAIA probe has some modest advantages over existing chemical probes, the overall impact is modest and the scientific advances are incremental.

Thanks for this comment and we would like to further highlight the significance of our study. We do not agree that the overall impact of our work is modest. This is because currently there is no good acrylamide-containing cysteine probe for cysteine profiling, and this should be critical because of preferential identification of lead compounds from probe containing the same reactive warhead. **In view of the fact that many bioactive and clinically used cysteine-targeting covalent drugs are acrylamide compounds, development of promising acrylamide probe should be of high significance (as supported by the comment from reviewer 2).** In our study, we report a versatile strategy, using electron-deficient nitrogen, to build an activated acrylamide to improve its Cys reactivity. NAIA, as an example of probe developed from this strategy, has been found to demonstrate superior performance on cysteine profiling to identify new ligandable hotspots, including those on cancer drivers that are potential hotspots for therapy (e.g. POLRMT and RAD21; Figs. 8d and 8e). We further demonstrate the successful application of NAIA in covalent ligand screening experiment to discover CL1, which is a lead compound that can mediate G₁-phase cell cycle arrest by covalent targeting of Rac1, as supported by flow cytometry, western blotting, biochemical assays and genetic knockdown experiments (Fig. 9).

In addition to profile functional and ligandable cysteines, we also illustrate the potential application of NAIA as a molecular probe for imaging cellular thiol oxidation by confocal fluorescence microscopy, which has not been realized by conventional cysteine probes. In the revised manuscript, we have further employed NAI/NAIA-5 couple to identify oxidized cysteines by MS experiment (Fig. 6 in the revised manuscript). We successfully profile unique and important cysteines associated with redox biology

which have not been reported by the CPT platform. Interestingly, we found very different oxidation profiles of cysteines on important classes of proteins such as thioredoxins and glutathione peroxidases, as well as proteins associated with hydrogen peroxide metabolism. This suggests different modifications and sensitivity toward oxidation of these cysteines. All these results illustrate the good potentials of NAI/NAIA for studying thiol reactivity and complex cysteine biology by microscopy imaging and MS experiments.

With all these advancements and comprehensive data spanning from chemistry and chemoproteomics to biology and covalent ligand discovery, the present work should bring good impacts on probe development for chemoproteomics and molecular imaging, as well as research on redox biology and covalent ligands for therapy.

In addition, key experiments and comparisons are not performed.

We thank so much for this suggestion so we can perform more experiments to further strengthen the quality of our work. In addition, to make the significance and impact of our work more visible to the readers, we have discussed in more details our key findings in the discussion section (p.13-15).

1. The primary advantage of NAIA that the authors focus on is reactivity. However, the authors compare NAIA to iodoacetamide, despite it is being well known that NEM is a much faster and more specific reagent than iodoacetamide (Analytical Biochemistry 358 (2006) 171–184, <https://doi.org/10.1016/j.bbapap.2012.08.002> and many more). There is also a clickable, alkyne version of NEM commercially available. A comparison to NEM is needed to evaluate the improvement in reaction kinetics of NAIA to contemporary cysteine labeling reagents.

We thank so much for this suggestion. We agree that maleimide is another important chemical moiety that has been employed for Cys profiling. In the original manuscript, we picked iodoacetamide (IA)-based probe for comparison because of its widely use for profiling functional cysteines (some examples: Nature 468, 790–795 (2010), Nature 534, 570–574 (2016), Nat. Biotechnol. 39, 630–641 (2021) and Nat. Commun. 12, 1415 (2021)) as well as for probing drug target in covalent drug development (e.g. KRAS G12C inhibitors: Cell 172, 578-589.e17 (2018) and Nature, 575(7781), 217–223 (2019)). For maleimide, its cysteine adduct is known to be unstable (due to retro-Michael addition) and undergoes hydrolysis readily (Nat Biotechnol 30, 184–189 (2012) and ChemBioChem 17, 529 (2016)), and these are

probably the reasons why maleimide-based probe is relatively less popular than IA-based probes for cysteine profiling.

We agree that comparison of NAIA and maleimide-based probe, MIA, should be interesting as MIA is known to have very fast cysteine reaction kinetics. **In this revision, we have finished LC-MS analysis of reactivity with Cys or Glutathione (Figs 2c, 2d, S4 and S5), as well as gel-based ABPP experiments to investigate the ability of the probes to capture functional cysteines in cell lysates and live cells (Fig. 3).** We found that MIA reacts slightly faster than NAIA (Figs. 2c and 2d), but it was quickly consumed by glutathione (GSH) which is the most abundant cellular thiol (Fig. S4). Unlike MIA, NAIA showed fast reaction with Cys while reacted significantly slower with GSH (Fig. S5). The slower reaction of NAIA with GSH is probably due to its lower preference in labeling Cys on peptides with a local sequence of GCD (GSH is a short GCD tripeptides), as found in MS experiment of cysteine profiling in live HepG2 cells (Figs. 7h and S19). **This allows NAIA to capture more functional cysteines in cell lysates and live cells than MIA at all the tested incubation time points and concentrations in the gel-based ABPP experiments (Fig. 3). This highlights the superior performance of NAIA for profiling functional cysteines, in both cell lysates and live cells, over maleimide-based probe.**

We would also like to emphasize that **the focus of the present study is not just on Cys reactivity, but is the Cys reactivity of acrylamide compound so that a more promising acrylamide-containing Cys probe can be developed.** This is particularly important because many cysteine-targeting covalent drugs are found to be acrylamide compounds. Preferential identification of drug lead compounds from probe with the same reactive warhead has been reported, while currently there is no acrylamide-containing probe with satisfactory Cys reactivity for Cys profiling. **As a result, our study should provide good insights into activating acrylamide warhead to achieve higher Cys reactivity and hence to develop promising acrylamide-containing cysteine probe. This should facilitate research on cysteine-reactive compounds for translational and clinical applications.**

2. The authors claim that NAIA can uncover a ‘unique pool of cysteines that can be liganded as new drug opportunities. However, they make this claim based on comparison to labeling with IAA at a concentration they admit is suboptimal. To claim that these sites are uniquely labeled by NAIA requires comparison to optimal levels of IAA or other studies that have used IAA-based warheads for redox proteomics such as CPT6 (DOI: 10.1016/j.cell.2020.02.012)

and other studies employing IAA-alkyne (<https://doi.org/10.1016/j.cell.2017.08.051>, <https://doi.org/10.1021/acscchembio.9b00424>).

Thanks so much for this suggestion. In the original manuscript we have already compared the cysteines and proteins captured by **NAIA with the results in the current state-of-the-art profiling experiment by DBIA (Fig. 7d in the original manuscript) and the cysteine chemoproteomics database CysDB (text, p. 9 and 12) from reported cysteine probes at optimal conditions.** Our NAIA, at 10 μ M working concentration, can identify a significant amount of new functional cysteines and proteins that have never been found by reported probes at optimized conditions. In the revised manuscript, **we further conducted cysteine profiling experiments in live HepG2 cells using NAIA-5, and profiled more new functional cysteines (Figs. 7i and 7j). We also show that these functional cysteines are ligandable by competitive MS-based ABPP experiment using CL-Sc as scout compound (Fig. 8). Our result reveals that >600 ligandable cysteines identified by NAIA-5 were not found in the reported study of DBIA (Nat. Biotechnol., 2021), and many of them are hotspots for therapy, including those on cancer drivers such as POLRMT and RAD21 (Figs. 8d and 8e).**

In addition, as suggested by this reviewer, comparison to other reported cysteine profiling experiments using IAA under optimal conditions (*Cell* 171, 696-709.e23 (2017)) or with advance chemically cleavable linker (*ACS Chem. Biol.* 14, 1940–1950 (2019)) also revealed identifications of a large population of functional cysteines (>2,700) by NAIA-5 (Fig. S18).

In order to compare with CPT platform which is about redox proteomics, we have conducted new MS experiment, using NAI/NAIA-5 couple to identify oxidized cysteines (Fig. 6 in the revised manuscript). ~ 30,000 NAIA-5-labeled peptides on average were detected in HepG2 cells treated with solvent vehicle, H₂O₂ and H₂O₂+NAC (184,644 modified peptides in aggregate; Figure 6b and Supplementary Data 1), with a total of **12,584 cysteines being identified. Among these cysteines, 5,625 cysteines have not been reported by the powerful cysteine-reactive phosphate tag (CPT) platform for redox proteomics (Fig. S14), highlighting the robust application of NAIA-5 as a cysteine probe for MS experiment.**

Also, we compared the 696 cysteines with changes in activity under H₂O₂ treatment with the reported redox proteomics results from CPT. Our NAI/NAIA-5 couple enables profiling 338 unique redox-sensitive cysteines that were not reported by CPT (Figure 6f).

All the above results confirm that our NAIA-5 enable identifications of new functional (Figs. 6f, 7d, 7i, S14 and S18) and ligandable (Fig. 8b) cysteines which have not been reported previously by other cysteines probes at their best conditions.

3. The authors note that a problem of IAA is off target reactivity with cysteine sulfenic acids, but do not evaluate the reactivity of NAIA with this group. Also, many off target effects of IAA were identified using unbiased proteomics and detecting other residues were modified. The authors did not consider modifications of other residues by NAIA in their proteomic data analysis and thus might have missed off target labeling.

We thank a lot for this suggestion on using unbiased open search to identify off target labeling of NAIA onto other amino acids as well as sulfenic acid. Previously we have data on selectivity of NAIA for amino acid in the MS experiment by targeted search for modifications (Fig. 7b in the original manuscript). **Now the selectivity data of NAIA-5 from the MS experiment was updated with the results obtained by unbiased open search (Figs. 7b and S17 in the revised manuscript). NAIA was found to show excellent selectivity, as compared to IAA, toward cysteines but not other amino acids or sulfenic acid.**

4. While the cell cycle arrest by CL1 is interesting, binding competition experiments are now commonplace and the data provide no mechanistic understanding of how CL1 works. What is the functional target, or what is the mechanistic link to cell cycle proteins?

We thank so much for this comment and agree that it is important to understand the mechanism of action of **CL1**. From the protein targets of **CL1**, we found that Rac1 is a known cancer driver and has been reported to modulate cell cycle. This prompted us to investigate the effect of **CL1** on Rac1 biology through covalent targeting the Cys178.. **We first confirmed *in vitro* binding of CL1 with purified human Rac1 protein by gel-based ABPP experiment (Fig. S23). Then, CL1 was found to disrupt *in vitro* interactions between Rac1 and TIAM1, which is important for activation of Rac1 (Fig. 9f). In consistent to this result, live HepG2 cells incubated with CL1 showed a significant decrease in Rac1-GTP level, which is the active form of Rac1 (Fig. 9g), as well as a decrease in pPAK which is the downstream signal of Rac1 (Fig. 9h). Confocal fluorescence imaging using FRET-Rac1 biosensor also revealed a decrease in Rac1 activity upon CL1 treatment (Fig. 9i). In addition, CL1 treatment led to decreases in pRb, E2F1 and cyclin D1 levels in HepG2 cells (Fig. 9j),**

indicating cell cycle arrest at G1 phase. Genetic knockdown of Rac1 rescued HepG2 cells from cell cycle arrest, as shown in flow cytometry (Figures 9k and 9l) and western blotting experiments (Fig. 9m). This supports the effects of CL1 on cell cycle in HepG2 cells are primarily through covalent targeting of Rac1.

5. The authors claim that “Trapping free thiols in live cells should be preferable” to in vitro labeling. But, it is well known that cysteine alkylation reagents are highly toxic, with general agreement that blocking critical thiols in enzymes will disrupt cell function, blocking cysteines important for antioxidant capacity will induce oxidative stress which – in turn – will lead to cysteine oxidation that will prevent NAI/NAIA from binding. In short, the debate between live cell labeling and in vitro labeling is nuanced, complex, and discussed in many reviews and opinion pieces and it is not self-evident that trapping free thiols in live cells is preferable.

We thank the reviewer for this constructive comment. In the original manuscript we mentioned that “Trapping free thiols in live cells should be preferable” in view of the previous sentence “highly dynamic nature of thiol oxidative modifications may result in loss of these modifications throughout cell lysis process in OxICAT”, **i.e. trapping free thiols should be preferable to prevent loss of oxidative modifications. Yet, we were not proposing that trapping free thiol in live cells is always the preferable approach. Sorry for any confusion caused, and we have revised the text in the revised manuscript to clarify our meaning (p.6, lines 158-160) that trapping free thiols in live cells can be a good alternative approach to study redox biology.**

In the revised manuscript we have demonstrated that live-cell trapping of free thiol using NAI/NAIA couple enables successful profiling of oxidized cysteines in cells incubated with H₂O₂ (Fig. 6). A large number of unique and important cysteines associated with redox biology are profiled in our experiments, which have not been reported by the powerful CPT platform for redox proteomics. This suggests the good values of our NAI/NAIA couple for studying redox biology on these newly profiled cysteines.

We totally agree that no methodology/platform is perfect for every experiment, so this is the motivation for us to develop new tools that allow live-cell trapping of free thiols to study redox biology, which has not been realized so far. The promising results from NAI/NAIA couple to image and profile oxidized cysteines suggest that NAI/NAIA have good

values to study redox biology, and in principle can work complementary to the established platform such as OxICAT to get a more conclusive result on cysteine oxidative modifications.

It is noteworthy that NAI/NAIA couple can work for trapping free thiols in cell lysates as well. In view of the superior cysteine reactivity of NAI as compared to the conventional IA-based trapping agent, the utilization of NAI in OxICAT as trapping agent may help to avoid loss in dynamic and unstable cysteine oxidative modifications throughout cell lysis.

It is notably that while NAIA is more reactive than IAA it is shown to be less toxic in Fig 5C. Can the authors reconcile this paradox?

Thanks so much for this comment. We found that, unlike MIA, NAIAs reacted much slower with GSH as compared to its reaction with Cys in LC-MS experiments (Fig. S5). **Based on the analysis of the MS data from cysteine profiling experiments in live HepG2 cells (Figs. 7h-7j), we found that NAIA-5 has lower preference in labeling cysteines on peptides with GCD local sequence than IAA (Fig. S19). This could explain the relatively low reactivity of NAIA-5 toward GSH which is a tripeptide of GCD. Since GSH is an important cellular anti-oxidant and changes in its activity should lead to oxidative stress and cell death, the higher relative reactivity of IAA with GSH than cysteines on other proteins should be one of the main reasons accounting for its higher cellular toxicity than NAIA.**

Reviewer #2:

In their manuscript, Lei et al. Have developed N-acryloylindole-alkynes (NAIAs) as promising new probes for the proteome-wide profiling of cysteine residues, a technology that has vast implications for understanding fundamental biology and in drug discovery. Let me start by saying, that I am normally skeptical, if we need more probes for cysteine monitoring in these technologies as many of these probes already exists. Nevertheless, this work stands out in various aspects: 1) The authors show advantages of the new probe in various scenarios, not only over traditionally used iodoacetamide alkyne (IAA), but also over some other of the newest generation of probes. 2) The authors convincingly show applicability of the probes in living cells for microscopy and proteomics with low toxicity, which is very hard to achieve so far. 3) Maybe most importantly, the probe (to my knowledge for the first time in such a broad profiling probe) uses a terminal acrylamide as the reactive group. This has important implications, as acrylamides are the reactive group used mostly in clinically approved drugs and, therefore, being able to profile this compound class with a probe with the same reactive group could have huge implications for drug discovery.

We thank this reviewer for the strong endorsement of our work and positive comment on the significance and potential impacts of the present study.

I, here, agree with the authors that state: “It is noteworthy that currently there is no promising acrylamide-containing probe developed for cysteine profiling, while many cysteine-reactive therapeutic agents with good bioactivity and efficacy are indeed acrylamide-containing compounds. Successful development of a good acrylamide-containing cysteine probe, such as NAIA, is anticipated to facilitate research on cysteine-reactive compounds for translational and clinical applications.”. I am, therefore, convinced that this probe will find a lot of important applications in chemoproteomics and is interesting to the broad readership of Nature Communications.

The research is performed to very high standards and covers a lot of different technologies. The authors synthesize the probes, characterize them with isolated amino acids as well as in vitro and in cells both in gel-based experiments, in MS-based proteomics and in microscopy studies. They perform clear and fair comparisons to state-of-the-art technologies to show where

their probes are superior. The authors make all proteomics data available through the PRIDE repository, which will enable others to quickly explore the power of the technology.

I am convinced, that this is an excellent study that should be published in Nature Communications. I only have a few minor comments that should be considered.

We thank the reviewer for the positive opinions on the quality of our work, as well as his/her support for our publication in Nature Communications. We also thank the reviewer for the constructive suggestions so that we can further strengthen the quality of this manuscript.

1) The authors throughout the paper define any cysteine they detect as “ligandable”. Nevertheless, the field and most prominently Ben Cravatt’s group has defined a ligandable cysteine to be engaged by a competitor at high stoichiometry (usually at least 75%). This makes a difference as the here detected cysteines can be engaged only to a low degree by NAIA, which is fine for detection, but does not allow to infer that they can also easily be made amenable to covalent drug discovery. In that sense, the authors in my eyes only identify 79 ligandable cysteines here (the ones strongly competed by CL1). This should not take away from the study, but I am convinced that this wording must be harmonized with what is standard in the field.

We thank the reviewer for this valuable suggestion. **Now we have performed a competitive MS-based ABPP experiments using a scout compound, CL-Sc (Fig. 8 and Supplementary Data 7), and successfully identified >1,100 ligandable cysteines based on a threshold of 4 for LFQ ratio which indicates 75% Cys occupancy. Over 50% of these ligandable cysteines have not been identified in the reported work of DBIA (Nat. Biotechnol. 39, 630–641 (2021)), and a number of them are located on cancer drivers, with POLRMT and RAD21 as notably examples.** Interestingly, the newly identified Cys on POLRMT (Cys591) and RAD21 (Cys392) are located close to the promoter-binding domain and at the binding interface of RAD21 and SA2, respectively. Therefore, covalent targeting of these Cys should allow modulation of activities of POLRMT and RAD21. These highlight the potential of NAIA to identify new hotspots for therapy.

2) The authors should provide detail, how the amino acid selectivity by proteomics has been determined.

We thank a lot for this suggestion. In the original manuscript, we performed targeted search for modifications on different amino acids (Fig. 7b in original manuscript). With the valuable comments from all the three reviewers, **we now have performed unbiased open search by MSFragger (Figs. 7b and S17 in the revised manuscript). NAIA was found to show excellent selectivity, as compared to IAA, toward cysteines but not other amino acids or sulfenic acid. More details about the data analysis can be found in the Method section (p.20).**

3) On page 3: “Yet, low cysteine reactivity has been found in aqueous buffer solution for acrylamide compounds, similar to iodoacetamides.” The reactivity of normal acrylamides should be much lower than iodoacetamide’s, right? Maybe “in comparison to” instead of “similar to”? I see the point that the reactivity of NAIA is still higher than iodoacetamide, but iodoacetamide and “normal” acrylamide are surely not similar in reactivity.

We thank a lot for this comment, and have revised the text accordingly (p.3, lines 54-55).

4) On page 10, the authors use competitive gel-based ABPP to prioritize compounds for MS-based proteomics. They write: “Interestingly, NAIA-5 enables identification of a number of covalent ligands showing good bindings with proteins in HepG2 cell lysates, as indicated by the decrease in in-gel fluorescence intensity as compared to the DMSO control (Figure 8a). On the other hand, IAA also identified some hit compounds, but fewer than those by NAIA-5.” It is currently very hard to see, in the gel, what they mean here. I think bands that are considered as liganding events should be marked. Also, as they make quantitative claims from these gels, how is a hit compound clearly defined here? I surely see the value of these experiments, but as only few cysteines can truly be differentiated on a gel, I would suggest to be careful with such quantitative claims.

We thank the reviewer for this important comment, and in the revised manuscript we have shown the quantification data of the competitive gel-based ABPP experiment (Fig. S21). **We measured fluorescence intensity of several protein bands, and those compounds with fluorescence intensity lower than 70% of that of DMSO control in 3 or more bands out of the 5 selected bands are considered as hit compounds.** The figure has been moved to

Supplementary Information because there are quite a number of new and important figures in the revised manuscript.

Reviewer #3:

Lai and colleagues introduce an optimized class of acrylamide-based reactive cysteine probes for in situ cell labeling. These probes are demonstrated to outcompete IAA by multiple experiments in human cell cultures at low working concentrations (10-50 μM) and its potential for covalent ligand discovery was also presented. Overall, the results are convincing, although some sections could be shortened and presented in a more intuitive way. I outline additional comments and concerns below. Nevertheless, I find these optimized trapping versions of acrylamide of interest to the general redox community.

We thank this reviewer for the positive comment on our work. We also thank so much for the constructive suggestions so that we can further strengthen the quality of this manuscript and present it in a more intuitive way.

Major:

- Proteomics:

o In the proteomic searches you specified cysteine carbamidomethylation as a fixed modification and then on top of that the variable modifications of NAIA-5 and IAM, I assume you then accounted for the fact that the total monoisotopic mass of the adduct should be reduced by 57 Da due to the fixed IAM setting?

We thank this reviewer for the comment. We did take into account the fixed IAM on Cys in the data searching, so the variable modifications for NAIA-5 and IAA are +681.38500 and +494.32167 respectively. More details about MS data analysis can be found in the Method Section (p.19-21)

o How did you perform the off-target labeling? You then thus conducted a blind search? Thus performing a mass tolerant search, e.g. in your case [-150 Da;+1000Da] for instance? Personally, when introducing new chemoproteomic probes, such searches are very informative and offer you an unbiased view of your introduced adducts and their location. Currently, I am unsure if this was performed here.

We thank the reviewer for this important comment. In the original manuscript, we performed targeted search for modifications on different amino acids (Fig. 7b in original manuscript) and did not conduct a blind search. With the valuable comments from all the three reviewers, **we now have performed unbiased open search by MSFragger (Figs. 7b and S17 in the revised manuscript). NAIA was found to show excellent selectivity, as compared to IAA, toward cysteines but not other amino acids or sulfenic acid. More details about the data analysis can be found in the Method section (p.20).**

o Did the labeled peptides contain any diagnostic fragment ions? Thus certain peaks resulting due to fragmentation of the relatively large NAIA group? If so, this could aid in confident identification of labeled sites.

We thank the reviewer for this constructive comment. We did perform a liable search on the MS data from HepG2 cell lysates labeled by NAIA-5. **With reference to the reported study (*BioRxiv*, doi.org/10.1101/2022.10.12.511963) and by using MSFragger, we identified one key fragment ion from NAIA-5-modified peptides, with delta mass of +3.9822. This should originate from ring-opening of indole to kynurenin in the fragment ion.**

In the revised manuscript, we did not consider this fragment ion in searching the MS data (Figs. 6-9). Therefore, we prefer not to mention this fragment modification in the main text in order to prevent confusions on the data analysis.

o Why was the chemoproteomics not attempted after labeling in live cells? This would give an even better result compared to IAM labeling and showcase the benefits of your probe?

We thank the reviewer for this valuable suggestion. Now **we have performed cysteine profiling experiments on live HepG2 cells treated with NAIA-5 or IAA (Figs. 7h-7j). Similar to the profiling experiment of HepG2 cell lysates, NAIA-5 enables identification of more functional cysteines than IAA. We have also compared the results of cysteine profiling experiments in cell lysates and live cells, and found a significant number of ligandable cysteines (1,384) that can be identified in live-cell experiment only (Fig. 7i). These cysteines identified in live-cell experiment only are less profiled previously, as compared to the cysteines identified in both cell-lysate and live-cell experiments. Furthermore, the proteins identified in live cell experiment only have a larger population**

of non-DrugBank proteins, as well as having different distribution of protein class (Fig. 7j). All these results suggest the presence of highly reactive cysteines in live cells, and they might lose their reactivity after cell lysis and hence they cannot be identified by MS experiment on cell lysates. Therefore, the superior performance of NAIA in live cell experiments should aid in discovering a unique pool of functional cysteines.

o I don't know whether isotopic labeling strategies with such probes could be discussed as a future option as well.

We thank the reviewer for this suggestion. We are currently developing next-generation NAI-based probes for cysteine profiling. Preliminary results show good compatibility of NAI-based probes with isobaric labeling such as TMT for quantitative analysis. **This has been mentioned briefly in the discussion section (p.14).**

- Page 5-6, Can we conclude already here from these fluorescence intensity that there are proteins uniquely labeled? The intensities are of course greater, and likely there will always be unique labeled proteins to either IAM or NAIA-4/5, however chemoproteomic approaches will only deliver you real quantitative and conclusive data to support these statements. Of course the greater band intensities seem to suggest that they might target additional cysteine targets.

We thank the reviewer for this comment, and agree that MS experiments will deliver the most conclusive data of unique cysteines and proteins by NAIA. In the original manuscript, we mentioned that in gel-based ABPP experiments, some fluorescence bands (not just intensity but also the labeling pattern) were found in sample incubated with NAIA only but not IAA (Figs 4a-4c in the original manuscript). This is the reason for us to suggest unique labeling of proteins by NAIA, which are later supported by the conclusive MS data. Yet, after receiving comment from this reviewer, **we agree that this could be better for us not to mention uniquely labeled proteins by NAIA based on the gel-based experiments. Therefore, we have removed the text about “unique labeling of proteins by NAIA” in the section describing the gel results.**

- Page 7, Figure 6b. I am confused by the IAM vs NAIA-5 confocal imaging. A similar comparison was done in the previous paragraph in Figure 5 and there 10 μ M IAA for 15 min

(Figure 5a) did give fluorescence, and now 50 μ M IAA for 10 min (Figure 6b) did not give any fluorescence? This is due to different imaging parameters, or I overlooked something?

We thank the reviewer for this comment. This is because of the different imaging parameters for acquiring Figs. 5a and 6b in the original manuscript. For Fig. 6b in the original manuscript (now Fig. 5b in the revised manuscript), HepG2 cells incubated with NAIA-5 at 50 μ M for 10 min were found to be super bright. Therefore, the laser power and voltage gain of the detector have to be set at much lower values in order to prevent saturation of the fluorescence from NAIA-5-labeled cells. Using these imaging parameters, only very weak fluorescence can be found from the IAA-labeled cells, as shown in Fig. 5b in the revised manuscript.

- As also touched upon in the discussion, I was wondering regarding the reactivity towards non-proteinaceous thiols. Could glutathione not be tested in vitro next to single amino acid residues like in terms of probe reactivity as in Figure 3e? You could then measure the probe consumption. In addition, confocal imaging experiments can be envisioned after perturbing the glutathione levels.

We thank the reviewer for this valuable suggestion. **We have conducted LC-MS experiments to investigate reactivity of NAIA-5 with GSH (Fig. S5). Unlike MIA, NAIA5 reacted much slower with GSH as compared to its reaction with Cys in LC-MS experiments. Confocal fluorescence imaging shows no significant changes of NAIA5 in labeling functional cysteines in cells with higher glutathione level (Fig. S12).**

We interpret the above results by using MS data from cysteine profiling experiments in live HepG2 cells (Figs. 7h-7j). We found that NAIA-5 has lower preference in labeling cysteines on peptides with GCD local sequence than IAA (Fig. S19). This could explain the relatively lower reactivity of NAIA-5 toward GSH which is a tripeptide of GCD, as well as no significant changes in capturing functional cysteines in cells with high GSH level in the confocal imaging experiment.

Minor:

- The manuscript would benefit from English proofreading.

- References often tend to be included as ‘.3,10–12’, should be ‘3,10–12.’?

- Page 3, liganded by activity-based probes

We thank the reviewer for these comments, and have revised the manuscript accordingly.

- Page 3, I guess it is logical activity-based probe are a critical component in activity-based protein profiling.. Sentence could be better rephrased to give the core message that development of activity-boost profiling could aid in targeting novel cysteine ligands and new druggable hotspots.

We thank the reviewer for this comment, and have rephrased the sentence (p.3, lines 37-39).

- Page 3, ‘by forming the acrylamide group using scaffold with less electron-rich nitrogen’, has to be rephrased

We thank the reviewer for this comment, and have rephrased the sentence (p.3, lines 57-60).

- Page 4, The first paragraph reads to me a bit too much like a Methods section, I think some details regarding the synthesis could be omitted and a reference could be made to the Methods section (or in this case Supporting information). Abbreviations such as ‘NAine’ are not introduced in the main text (was spelled out in the figure).

We thank the reviewer for this comment. We have shortened this paragraph, and at the same time introduced abbreviations including NAine (p.4, lines 82-90).

- Figure 2a, the reaction scheme is a bit confusing due to the ‘DDQ, toluene’ shift from right to left of the figure (i.e. from compounds 3a/3b to 4a/4b). Typically, you would go down and then left to keep a consecutive order.

We thank the reviewer for this comment, and have revised the schematic scheme accordingly (now Fig. 1b).

- Page 5, why refer to Figure 1c (introduction figure) regarding the reaction of NAIA-5 with N-acetylcysteine methyl ester?

We thank the reviewer for picking this typo, and have revised the text accordingly.

- Figure 3:

o Why were not the same time intervals could used for the SIC for substrate/adduct and LC-MS? Why was the LC-MS done at 5.6 min (= 336 seconds) and 6.2 min (= 372 seconds)? For instance, in Figures S1 and S2 the SIC time intervals were identical. Of course, the reaction and thus adduct formation of NAIA-5 is convincing.

We thank the reviewer for this comment. We have revised the LC-MS figure, showing the consumption of NAIA-5 and the formation of adduct at the same time intervals (now Fig. 2b).

The LC-MS was not done at 5.6 or 6.2 min. This refers to the MS recorded at $R_f = 6.2$ and 5.6 min, i.e. the MS of NAIA-5 and its cysteine adduct. They are to confirm the identity of NAIA-5 and the adduct eluted at 6.2 and 5.6 min respectively.

o It could be useful for visual interpretation to have a horizontal line at 10 μM , which should be the starting concentration/no reaction. Would also apply for Figure S1d.

We thank the reviewer for this comment, and have added the dotted horizontal line at 10 μM (now Figs. 2c and S1c).

o Why term it as a NAIA-5+Na⁺ adduct compared to NAIA-4+H⁺ adduct in Figure S1? The LC-MS peak intensities look identical, with the Na⁺ adduct being lower in intensity compared to the H⁺ adduct?

We thank the reviewer for this comment, and have revised the figures accordingly. Now we show the NAIA-5+H⁺ adduct in Fig. 2b.

- Would the timing of the complete reaction be more accurately described by overlaying Figure S1c (NAIA-4) and Figure 3c (lower panel, NAIA-5)? Now it says the reaction of NAIA-5 was complete within 300 s, but judging from Figure 3c it happens around ~ 120 s?

We thank the reviewer for this suggestion, and have prepared a new Fig. 2c (Fig. 3c in the original manuscript) showing the overlay plots of NAIA-4, NAIA-5, MIA, IAA and the NAine compound 3b. Also, now we describe the completion time of NAIA-5 more accurately in the main text, and it is within 120 s (p.4, line 93).

- Page 5, 'NAIA-5 and IAA were capable of labeling reactive cysteines on proteins in HepG2 cell lysates.' How did you confirm it? I was initially confused as I thought Figures 4a and 4b were already performed in live cells.

We thank the reviewer for this comment. The text in the original manuscript was describing the working principle of gel-based ABPP experiment to study cysteine labeling by NAIAs in cell lysates. We apologize for the confusion caused, and have revised the text to deliver a clearer message: "In the gel-based ABPP experiments, proteins in HepG2 cell lysates labeled by the probe would allow further conjugation to azide-fluor 545 through copper(I)-catalyzed alkyne-azide cycloaddition (CuAAC) and hence showed strong fluorescence..." (p.5, lines 118-119).

- Figure S1:

o There is an error in the legend: "The mass spectra (MS) at 5.19 and 4.82 min confirm the identity of NAIA-5 and adduct", this should be "NAIA-4 and adduct" I believe.

We thank the reviewer for picking this typo, and have revised the legend.

o Error bars missing for Figure S1d, or only one replicate (not indicated in legend)?

We thank the reviewer for this comment. This data in Fig. S1d is from a single experiment so there are no error bars.

- Figure S2: '[MBST] / μM ' for y-scale in panel c? Why ranging above the starting concentration of 10 μM ?

We thank the reviewer for this comment. After calibration, we found that there was small derivation of the starting concentration of the MBST (11.4 μM) from the expected concentration (10 μM). We have revised the legend and put down the exact concentration of MBST used in this experiment.

- Figure S3: the concentration of N-hydroxybenzimidoyl chloride can not be determined for panel c as before for Figure S1, S2 and Figure 3? Makes it difficult for a comparison of this graph to the other probes.

We thank the reviewer for this comment. Yes, we found that *N*-hydroxybenzimidoyl chloride did not ionize well in the ESI-MS. Therefore, the experimental setting for *N*-hydroxybenzimidoyl chloride was different from the other cysteine probes.

- Page 6, Although agreeing, I would prefer a more quantitative value than a ‘huge difference’ in in-gel fluorescence intensities between NAIA-5 and IAA labeling.

We thank the reviewer for this comment, and now have revised the text with descriptions of the quantitative change: “The in-gel fluorescence intensities from NAIA-5-treated cells surpassed those from IAA-treated cells (Figures 3d, 3g and S11), particularly at 15 min incubation (>3-fold difference at 10 μM).” (p.5, lines 129-131)

- Figure 4:

o I think Figure 4a is dispensable as Figure 4b is giving a more detailed overview of the labeling in cell lysates and indicating time ranges (which are lacking for Figure 4a). It would also shorten the text and give the main results in a more concise manner.

We thank the reviewer for this comment. Figure 4a in the original manuscript (now Fig. 3a) represents the results from HepG2 cells incubated with NAIA-5/IAA for 60 min, and it is different from Figure 4b (now Fig. 3b) which is from cells incubated for 15 or 30 min. We have now labeled Fig. 3a clearly the incubation time.

o Which statistical test was used to determine these p-values? Should be indicated in the legend.

We thank the reviewer for this comment, and have described the statistical analysis (unpaired two-tailed Student's t-test) in the legend of Figs. 3, 4, 5, 7 and 9.

- Page 6, I would not fast-forward in the text referring to Figure 5 in the first section. It makes more sense to give a conclusion at the end of the section that the differences can be attributable to both the uptake and enhanced reaction kinetics.

We thank the reviewer for this comment, and have re-ordered the sentences and put the conclusion sentence at the end of the paragraph (p.5-6, lines 131-137).

- Page 7, please introduce/spell out 'NAC'.

We thank the reviewer for this suggestion, and now we have introduced NAC on p.7, line 169.

- Figure 6:

o The legend for panel a reads set-up to monitor oxidative modifications, however that only applies to panels e and f in my opinion. Regarding panels a to d you look at reduced cellular labeling due to thiol oxidation, but you do not directly use NAIA-5 to monitor oxidized thiols (well you could say in an indirect manner).

We thank the reviewer for this suggestion, and have revised the legend for Fig. 5a in the revised manuscript as "the experimental setup to monitor levels of reduced cellular thiols by NAIA-5."

o How was NAC applied to the cell cultures? Pre-incubate together with H₂O₂ for 15 min or? The legend is not very descriptive for panel c and d.

We thank the reviewer for this suggestion. In view of the limitations of 350 words for the figure legend, we prefer to put down these experimental details in the Method section (p. 17, lines 542-543; also p.18, lines 559-560). The cells were incubated with a mixture of H₂O₂ (1 mM) and NAC (5 mM) in complete medium for 15 min at 37 °C.

REVIEWERS' COMMENTS

Reviewer #1 (Remarks to the Author):

With the revisions and new experiments presented, I feel this manuscript is now acceptable for publication.

Reviewer #2 (Remarks to the Author):

In their revision, the authors have convincingly addressed my minor comments. I am convinced that this manuscript is now ready for publication.

Reviewer #3 (Remarks to the Author):

In the revised manuscript, Koo et al has adequately addressed my previous concerns. The revised version is clearly written and well structured. In general, I found that this contribution is of great importance and interest to the community of redox biology.

I only have two minor comments:

1, In supplemental Fig.S17, panel b, I suggest the authors remove "Cys" from x-axis, it now looks like the % of modification for Cys is 5.

2, The authors did open search in MS analysis for Cys labeling by NAIA, but only checked the predicted mass increase. What about unpredicted mass increases that hint to unspecific labeling caused by other possibilities, e.g. the degradation of the chemical?

Response to Reviewers' comments:

Our point-by-point responses are in blue font.

REVIEWER COMMENTS**Reviewer #1:**

With the revisions and new experiments presented, I feel this manuscript is now acceptable for publication.

Thanks so much for all your comments and suggestions so that we can improve the quality of our paper. Thanks a lot for your support in the publication of our revised manuscript.

Reviewer #2:

In their revision, the authors have convincingly addressed my minor comments. I am convinced that this manuscript is now ready for publication.

Thanks so much for your endorsement of our work, and good suggestions so that we can strengthen our paper.

Reviewer #3:

In the revised manuscript, Koo et al has adequately addressed my previous concerns. The revised version is clearly written and well structured. In general, I found that this contribution is of great importance and interest to the community of redox biology.

Thanks so much for your suggestions so that we can strengthen our paper and present it in a well-structured way. We also thank for your endorsement of our work and positive comment on the importance of our study.

I only have two minor comments:

1, In supplemental Fig.S17, panel b, I suggest the authors remove "Cys" from x-axis, it now looks like the % of modification for Cys is 5.

We thank for this suggestion and have revised the figure accordingly.

2, The authors did open search in MS analysis for Cys labeling by NAIA, but only checked the predicted mass increase. What about unpredicted mass increases that hint to unspecific labeling caused by other possibilities, e.g. the degradation of the chemical?

We thank so much for this important comment. In addition to the expected mass shift due to cysteine modification by NAIA-triazole-DTB (the click reaction product of NAIA-5 and DTB-azide), we did observe some other mass shifts such as cysteine modification by NAIA-5 (without click reaction with DTB-azide) or by hydrolyzed product of NAIA (i.e. acrylic acid-cysteine adduct). This, together with the unique fragment ion (kynurenin) identified from labile search on the NAIA-modified peptides (thanks again for the good suggestion from this reviewer in the 1st revision), suggests that more functional/ligandable cysteines and proteins can be identified by NAIA if we can optimize our MS preparation protocol and data analysis pipeline. We are actively working on these two parts in order to further improve the performance of NAIA for cysteine profiling.

Nonetheless, in this paper we only utilized the expected mass shift from NAIA-triazole-DTB for the analysis of all different groups in our experiments. This should be a fair analysis and can lead to a valid conclusion in our study.

In order to provide more information about other possible mass shifts from NAIA labeling on cysteines, we now have added the open search result of HepG2 cell lysate profiled by NAIA-5 vs IAA to the Supplementary Data 2.